# TR-FRET between engineered nanobodies reveals the existence of endogenous CXCR4 oligomers
Joyce Heuninck[1,7], Vladimir Bobkov[2,3,7], Claire M. Grison[1], Amos Fumagalli[1], Mathias Lescuyer[1], Omolade Otun [1], Cherine Bechara[1], Laurent Lamarque[4], Eric Trinquet[4], Françoise Bachelerie[5], Philippe Marin[1], Bernard Mouillac [1], Sébastien Granier [1], Bas van der Woning[2], Hans de Haard[2], Martine J. Smit [3], Raimond Heukers [3,6] ✉ & Thierry Durroux [1] ✉

Chemokine receptors CXCR4 and ACKR3 are involved in multiple physiological and pathological processes. In overexpression systems, CXCR4 and ACKR3 form oligomers that are important for chemokine recognition and signaling. Whether this holds true in physiological settings is unknown as no evidence of such oligomers are available. To address this gap of knowledge, we develop a new approach for GPCR oligomer detection by time-resolved Förster resonance energy transfer (TR-FRET) between fluorescently labeled nanobodies. Fluorescent probes are conjugated to the C-terminus of extracellularly binding nanobodies using site-directed sortase-mediated labeling, retaining high affinity of the nanobodies for their target receptors. This approach, first validated in transfected cells by detecting the presence of CXCR4 and ACKR3 homo-oligomers and hetero-oligomers, reveals for the first time endogenous CXCR4 homo-oligomers in non-transfected human leukemia/lymphoma-derived cancer cell lines. More importantly, the investigation in peripheral blood mononuclear cells strongly supports the existence of CXCR4 oligomers in native conditions.

G protein-coupled receptors (GPCRs) are the largest family of cell surface signal transduction proteins. They are activated by numerous extracellular molecules, play a key role in cell-cell communication and regulate a variety of physiological and pathological processes. Hence, they also represent main targets for drug development[1]. Although GPCRs were shown to transduce signals in a monomeric form[2], multiple studies showed that GPCRs are organized as both homo- and hetero-oligomers associated with different signaling profiles[3,4]. GPCR oligomers can be formed within recently proposed 'hot spots', regions in plasma membranes where GPCRs and G proteins preferentially reside and interact to produce rapid and local signals[5]. Nevertheless, oligomerization of GPCRs has mostly been investigated in transfected cells[3,6,7]. The formation of oligomers in native tissues is still questionable since their existence has only been demonstrated in a few studies[8-10].

Chemokine receptors CXCR4 and ACKR3 (previously referred to as CXCR7) belong to the superfamily of GPCRs and play an important role in the immune system[11]. Their dysregulation, including overexpression, may result in many inflammatory and autoimmune disorders, as well as cancer[12]. ACKR3 has been reported to act as a scavenger of CXCL12, the endogenous chemokine ligand for both ACKR3 and CXCR4, resulting in a reduction of the CXCR4-mediated response[13]. The existence of CXCR4 homo-oligomers has been reported in heterologous systems by many studies using different technical approaches, including crystallography[14,15], bioluminescence[16,17] and fluorescence[18] resonance energy transfer (RET), single-molecule microscopy[19-21] and fluorescence fluctuation spectroscopy[21,22]. Disrupting CXCR4 homo-oligomers by a peptide corresponding to the transmembrane domain (TM) 4 of the receptor decreases the migratory capacity of tumor cells[18]. Similarly, a constitutive BRET signal between ACKR3 receptors has been reported[17,23]. Different studies also reported that co-expression of CXCR4 and ACKR3 has consequences on cellular signaling[13,17,24-28], suggesting a specific role of the receptor hetero-oligomers which were also shown to co-localize in microscopy[29]. Nevertheless, the importance of the oligomers in physiology is still questionable since the evidence of the direct interactions between these receptors in endogenous settings is still lacking.

[1]IGF, Univ Montpellier, CNRS, INSERM, Montpellier, France. [2]Argenx BVBA, Zwijnaarde, Belgium. [3]Division of Medicinal Chemistry, Amsterdam Institute for Molecules, Medicines and Systems (AIMMS), VU University Amsterdam, Amsterdam, The Netherlands. [4]Revvity, Parc Marcel Boiteux–BP 84175, Codolet, France. [5]Inserm UMR996–Inflammation, microbiome and immunosurveillance, Université Paris-Saclay, Clamart, France. [6]QVQ Holding BV, Utrecht, The Netherlands. [7]These authors contributed equally: Joyce Heuninck, Vladimir Bobkov. ✉e-mail: r.heukers@vu.nl; thierry.durroux@igf.cnrs.fr

Moreover, the direct link between the existence of oligomers and observation of specific signaling patterns remains to be established.

Studying GPCR oligomers in a native context is highly challenging. Direct interactions between receptors can be followed with RET-based strategies[30]. Yet, the fusion of fluorescent, bioluminescent or self-labeling proteins to receptors is not compatible with native conditions. We previously addressed this issue by time-resolved fluorescent RET (TR-FRET) using high affinity fluorescent antagonists to detect the receptor oligomers[8,10]. Interestingly, in these studies, fluorescent agonists developed a much smaller TR-FRET signal due to negative cooperative binding[8]. Therefore, fluorescent derivatives of CXCL12[31], which still retain agonistic properties, are not suitable to investigate CXCR4 and ACKR3 oligomerization. Among the other ligands available for CXCR4 and ACKR3[32,33], no high-affinity fluorescent small molecules have been developed so far.

One alternative involves the labeling of receptors with specific fluorescent antibodies. However, using conventional antibodies with the size of up to ~150 kDa, a RET signal can be observed even when receptors are at the distance of up to 50 nm and thus not in close proximity[34]. For that reason, we opted to detect chemokine receptor oligomers using nanobodies, which are single domain antibody fragments from camelid-derived heavy chain-only antibodies[35]. Nanobodies are highly selective, exhibit high affinity for their targets, and have the size of about 15 kDa[36], ensuring that an observed RET signal corresponds to a direct interaction between receptors.

To that end, we labeled CXCR4- and ACKR3-specific nanobodies with TR-FRET fluorescent donor and acceptor molecules in a site-directed and stoichiometry-controlled fashion using a method based on sortase A from *S. aureus*[37]. The labeled fluorescent nanobodies retained high affinities for the CXCR4 or ACKR3 receptors. This nanobody-based strategy to detect oligomers with TR-FRET was first validated in CXCR4- and/or ACKR3-transfected CHO-K1 and HEK293T cells. These two cell lines are compatible with most of the pharmacological and microscopic strategies. Moreover, they do not express endogenous ACKR3 (CHO-K1 and HEK293T) or CXCR4 (CHO-K1) or at a low level (HEK293T), preventing any impact of these endogenous receptors on the results. The strategy was further transferred onto detection of endogenous receptor oligomers expressed in various cancer cell lines, such as Raji, Jurkat, and CCRF-CEM cells. Finally, the strategy was implemented on human peripheral blood mononuclear cells (PBMC). The specific TR-FRET signal detected in these cells provided the first compelling evidence for the existence of CXCR4 homo-oligomers in non-transfected cells, indicating that their oligomerization does occur under native conditions and may be of physiological relevance.

## Results

### CXCR4- and ACKR3-specific nanobodies do not affect oligomerization of their target receptors

To demonstrate the existence of receptor oligomers in an endogenous setting, appropriate TR-FRET probes should not promote or disrupt oligomerization upon binding to the receptors. For that reason, we developed TR-FRET compatible fluorescently labeled nanobodies. They exhibit a size of ~10 times smaller than conventional antibodies and about two times larger than CXCL12 (Fig. S1). We used the previously characterized antagonistic CXCR4-targeting nanobody VUN400 (Fig. S2A)[38,39]. Using a similar approach[38], ACKR3-targeting nanobody VUN700 was developed (Fig. S2B). VUN700 exhibited a high affinity for ACKR3 in CXCL12 binding competition (Fig. S3A, $pK_i$ 7.9 ± 0.1, $n = 3$) and displayed antagonistic properties in CXCL12-induced β-arrestin2 recruitment (Fig. S3B and C, $pIC_{50}$ 7.5 ± 0.1, $n = 3$).

Next, we ascertained that the nanobodies themselves did not induce a change in the receptor oligomerization or the conformation of CXCR4 and ACKR3 homo-oligomers by following the BRET signal between the tagged receptors (Fig. S4A, B). By contrast to CXCL12, addition of VUN400 did not modify the BRET signal intensity (Fig. S4C). Similarly, addition of VUN700 did not modify BRET signal in ACKR3 expressing cells (Fig. S4D). Additional experiments were performed on HALO-CXCR4 in TR-FRET mode and neither CXCL12 nor VUN400 did affect TR-FRET ratio (Fig. S4E).

These results indicate that VUN400 and VUN700 at a concentration of up to 1 μM did not affect the oligomeric state nor the conformational rearrangement of the receptors. Therefore, both nanobodies appeared to be suitable tools to investigate the existence of oligomers in native conditions.

### Sortase-mediated fluorescent labeling retains high binding affinities of the nanobodies

Since the classical labeling methods based on random labeling of lysines can strongly reduce nanobody affinity, we opted for site-directed fluorescent labeling of VUN400 and VUN700 nanobodies using the sortase-based strategy (Fig. S2C). To allow the sortase-based labeling with the luminescent donor Lumi4-terbium (Lumi4-Tb) or the fluorophore acceptor d2, the VUN400 and VUN700 nanobodies were engineered to contain either a GGG- or LPETG-moiety on their N- or C-terminus, respectively (see Supplementary Information and Figs. S5-S9 for labeling strategies and chemical characterization of the labeled nanobodies). Binding affinities of the resultant fluorescently labeled VUN400 and VUN700 nanobodies were evaluated using Tag-lite assays[31] with N-terminally Halo- or SNAP-tagged CXCR4 or ACKR3 in transfected CHO-K1 cells. CHO-K1 cells were previously shown as not containing endogenous CXCR4 or ACKR3[40]. Kinetic binding experiments with N-terminally labeled d2-VUN400 (50 nM) showed no significant binding of the nanobody to the Halo-Lumi4-Tb labeled CXCR4 (Fig. S10A). By contrast, a significant TR-FRET signal for C-terminally labeled VUN400-d2 (50 nM) was detected and reached a plateau after 5 min (Fig. S10A). Saturation binding experiments with C-terminally labeled VUN400-Lumi4-Tb and VUN400-d2 confirmed that the nanobodies retained affinities in the nanomolar range for Halo-CXCR4 after labeling (Fig. S10B and Table S2). Similarly, VUN700-Lumi4-Tb and VUN700-d2 exhibited high affinities for SNAP-ACKR3-ΔCterm, a C-terminally truncated receptor devoid of internalization capacity[41] with Kd values of 19 ± 8 nM and 2 ± 0.2 nM, respectively (Fig. S10C and Table S2). Since CXCR4 and ACKR3 can bind CXCL12, we therefore investigated the selectivity of VUN400 and VUN700 by performing saturation experiments with VUN400-d2 and VUN700-d2 on both Lumi4-Tb-labelled HALO-CXCR4 and SNAP-ACKR3 (Fig. S11). No cross reactivity was observed and VUN400 and VUN700 exhibit a strong selectivity for their cognate receptor.

### Labeled VUN400 and VUN700 support the existence of CXCR4 and ACKR3 oligomers in transfected CHO-K1 cells

To detect the presence of CXCR4 oligomers using TR-FRET between the labeled nanobodies, CXCR4-transfected CHO-K1 cells were incubated in the presence of VUN400-Lumi4-Tb and VUN400-d2 at the same concentration. The signals of the donor, acceptor, and TR-FRET between the nanobodies were measured and plotted as saturation curves (Fig. 1A and Table 1). Similar experiments were performed with VUN700-Lumi4-Tb and VUN700-d2 using SNAP-ACKR3-ΔCterm transfected CHO-K1 cells (Fig. 1B and Table 1). Because the signals were saturable and the $TR-FRET_{50}$ values from both experiments (Table 1) are in the same range than the affinities determined using the Tag-lite binding assays (Table S2), it strongly suggests that TR-FRET results from the nanobodies bound onto their corresponding receptors. It is noteworthy that a rightward shift and sigmoidal curve could have been expected for the TR-FRET signal, but such a pattern may be difficult to discern if a large proportion of the receptors are dimeric, as has been reported for CXCR4 when expressed at high density[21] (see Supplementary Information – Binding curve modelling section and Fig. S18). Finally, similar results were obtained on cells co-expressing CXCR4 and ACKR3 receptors (Fig. S12).

In the competition experiments with a fixed concentration of VUN400-Lumi4-Tb (10 nM) and increasing concentrations of VUN400-d2 (ranging from 0.1 nM to 100 nM), the FRET signal followed a bell-shaped curve (Fig. 1C). The maximum signal corresponding to the equimolar labeling of CXCR4 with VUN400-Tb and VUN400-d2, is reached at 10 nM of VUN400-d2. This was expected considering that both labeled nanobodies have the same affinity. This strongly supports that the TR-signal corresponds to receptor oligomerization.

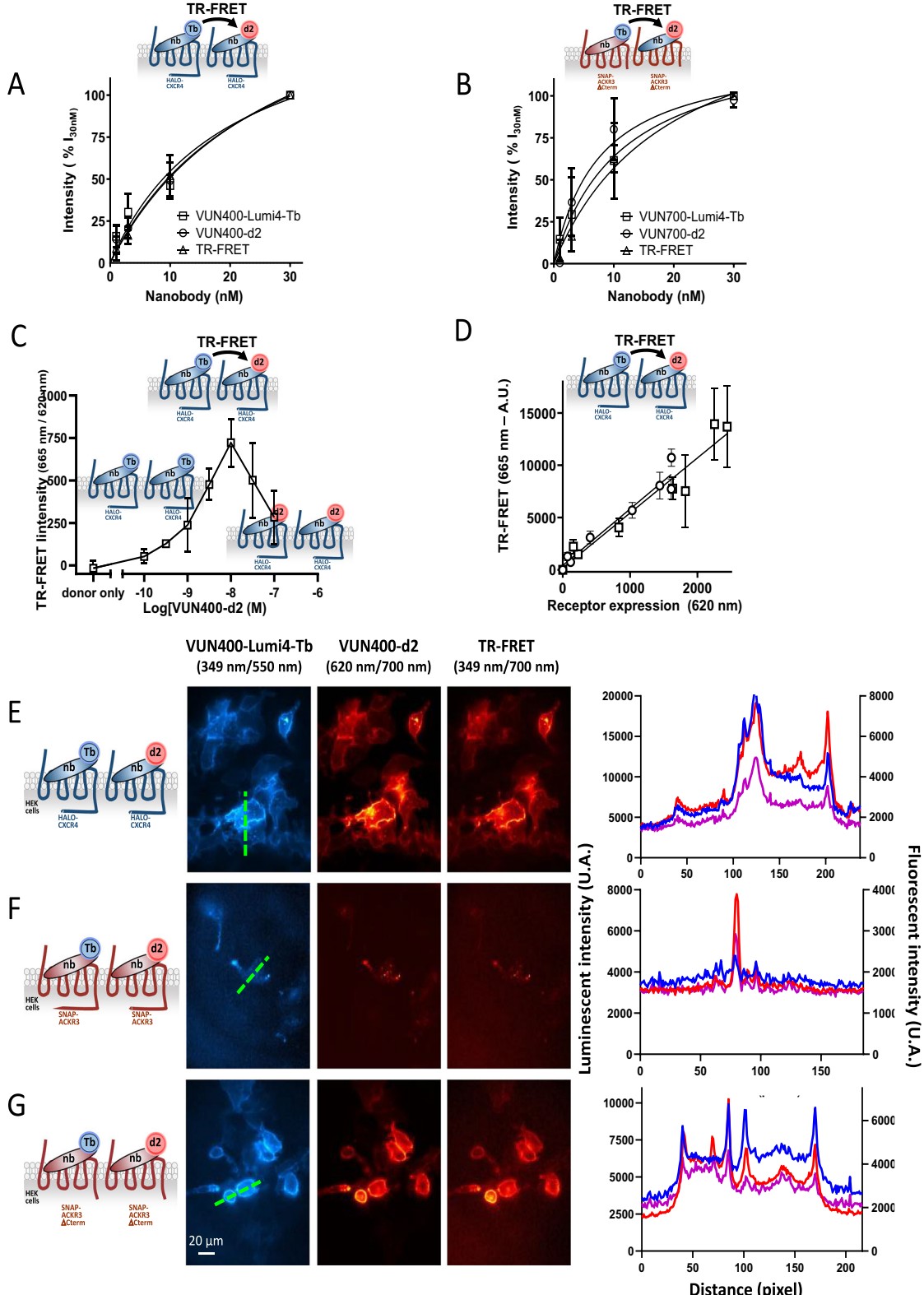

To exclude that the TR-FRET signal is only due to a dynamic TR-FRET, i.e., resulting from random collisions of receptors diffusing in the membrane, we carried out two types of experiments. First, we performed experiments on a panel of transfected CHO-K1 cells expressing increasing Halo-CXCR4 receptor densities and probed with VUN400-Lumi4-Tb and VUN400-d2. In parallel, the expression of Halo-CXCR4 was quantified by labeling Halo-CXCR4 with the Halo-Lumi4-Tb substrate. The TR-FRET as a function of the receptor density varied linearly (Fig. 1D, $R^2$ values > 0.8), by contrast to TR-FRET resulting from random collision (see Fig. S13). This indicates that TR-FRET efficiency within CXCR4 receptors is constant, not dependent on receptor density and therefore did not result from random collision of receptors diffusing in the membrane[42], strongly supporting the existence of CXCR4 homo-oligomers. In a second set of experiments, we used another d2-labeled nanobody targeting the mGluR2 receptor. We co-

**Fig. 1 | Detection of CXCR4 and ACKR3 homo-oligomers in transfected cells.**
**A** Association and TR-FRET curves for VUN400-Lumi4-Tb (squares) and
VUN400-d2 (circles) in Halo-CXCR4-expressing CHO-K1 cells. For TR-FRET
(triangles) both nanobodies were added to the cells at equal concentrations. All
values have been measured after performing washing steps to remove unbound
nanobodies. The specific binding signal presented here was calculated by subtracting
the non-specific binding signal, obtained in the presence of an excess IT1t from the
total binding signal. The emission bleed-through of Lumi4-Tb at the FRET wave-
length has been subtracted. Means ± SDs of values obtained in three independent
experiments are represented. **B** Association curves for VUN700-Lumi4-Tb
(squares) and VUN700-d2 (circles) in SNAP-ACKR3-ΔCter-expressing CHO-K1
cells. For TR-FRET (triangles), both nanobodies were added to the cells at equal
concentrations. All values have been measured under heterogenous conditions. Non
specific binding was determined in the presence of an excess of a mixture of unla-
beled ligand, VUN700, CXCL12 and VUF 15485. Specific binding was calculated by
subtracting the non-specific binding to the total binding. Means ± SDs of values
obtained in three independent experiments are represented. **C** Evolution of the TR-
FRET signal as a function of the donor/acceptor labeled nanobody ratio. CHO-K1
expressing CXCR4 were incubated in the presence of a 10 nM of VUN400-Tb and an
increasing concentration of VU400-d2 ranging from 0.1 nM to 100 nM. TR-FRET
signal was plotted as a function of VUN400-d2 concentration. Plotted data

correspond to means ± SDs from 3 independent experiments performed in tripli-
cate. **D** CXCR4 homomer detection in CHO-K1 cells with different CXCR4
expression levels. Cells were transfected with different concentrations of Halo-
CXCR4 cDNA and incubated with 20 nM VUN400-Lumi4-Tb and 20 nM
VUN400-d2. TR-FRET signals were plotted against the receptor densities, deter-
mined by incubation with Halo-Lumi4-Tb, and linear regression was determined.
These data points represent two independent experiments indicated with two dif-
ferent symbols (squares and circles). Error bars represent SDs of three technical
replicates for each data point. **E** CXCR4 homo-oligomers detected by TR-FRET
microscopy. HEK293T cells were transfected with Halo-CXCR4 plasmid. VUN400-
Lumi4-Tb and VUN400-d2 were added at a concentration of 10 nM for 2 h at 37 °C.
**F** ACKR3 homo-oligomers detected by TR-FRET microscopy. HEK293T cells were
transfected with SNAP-ACKR3 plasmid. VUN700-Lumi4-Tb and VUN700-d2
were added at a concentration of 10 nM for 2 h at 37 °C. **G** ACKR3-ΔCter homo-
oligomers detected by TR-FRET microscopy. HEK293T cells were transfected
with SNAP-ACKR3-ΔCterm plasmid. VUN700-Lumi4-Tb and VUN700-d2 were added
at a concentration of 10 nM for 2 h at 37 °C. Lumi4-Tb was excited at 349 nm and its
emission was detected at 550 nm. To acquire d2 images, samples were excited at
620 nm and emission was captured at 700 nm. To acquire TR-FRET images, the
excitation occurred at 349 nm and emission was detected at 700 nm. Scale
bar: 20 μm.

**Table 1 | Affinities of fluorescently labeled VUN400 and VUN700 for unlabeled CXCR4 or ACKR3**

| Nanobody pair | Detected signal[a] | p(TR-FRET$_{50}$) | | | | |
|---|---|---|---|---|---|---|
| | | Transfected CHO-K1 | Non-transfected cells with endogenous CXCR4 | | | |
| | | HALO-CXCR4 | Raji | Jurkat | CCRF-CEM | |
| VUN400-Lumi4-Tb + VUN400-d2 | Lumi4-Tb | 7.9 ± 0.52 | 8.1 ± 0.17 | 8.1 ± 0.11 | 8.3 ± 0.32 | |
| | d2 | 7.6 ± 0.09 | 8.1 ± 0.15 | 8.49 ± 0.09 | 8.8 ± 0.31 | |
| | TR-FRET | 7.5 ± 0.27 | 7.84 ± 0.24 | 8.1 ± 0.15 | 8.1 ± 0.13 | |
| | | SNAP-ACKR3-ΔCterm | Raji | Jurkat | CCRF-CEM | |
| VUN700-Lumi4-Tb + VUN700-d2 | Lumi4-Tb | 7.8 ± 0.62 | nd | nd | nd | |
| | d2 | 8.1 ± 0.23 | nd | nd | nd | |
| | TR-FRET | 7.7 ± 0.12 | nd | nd | nd | |

Cells were incubated in the presence of increasing equimolar concentrations of labeled VUN400 or VUN700 nanobody pairs. The signals of the donor, acceptor, and TR-FRET between the nanobodies were
measured. p(TR-FRET$_{50}$) values represented in the table are the means ± SDs values obtained in at least three independent experiments.
[a]Lumi4-Tb: excitation 337 nm, emission 620 nm; d2: excitation 645 nm, emission 680 nm; TR-FRET: excitation 337 nm, ratio 665/620; nd: not determined.

expressed the CXCR4 and mGluR2 receptors at comparable densities and
compared the TR-FRET signals obtained after incubation of CHO cells in
the presence of a mix of VUN400-Tb/ DN1-d2[43] or VUN400-Tb/VUN400-
d2 nanobodies. The results show a significant specific FRET signal with
VUN400-Tb/VUN400-d2 mixture as expected and a non-significant spe-
cific signal with VUN400-Tb/DN1-mGluR2 (Fig. S14). Again, the data
strongly suggest that the FRET signal does not result from random collisions
of membrane-diffusing receptors but from specific interactions between
CXCR4 receptors leading to the formation of homo-oligomers.

**Detection of CXCR4 and ACKR3 oligomers using labeled VUN400
and VUN700 in transfected HEK293T cells using TR-FRET
microscopy**
The labeled CXCR4 and ACKR3 nanobodies were then used to determine
localization of the receptors in HEK293T cells using TR-FRET microscopy.
HEK293T cells express only low levels of endogenous CXCR4 and no
detectable levels of ACKR3 (Fig. 3A). Therefore, HEK293T cells were
transfected with Halo-CXCR4, SNAP-ACKR3, or SNAP-ACKR3-ΔCterm
expression constructs and incubated in the presence of the donor- and
acceptor-labeled nanobodies and the signals of the donor, acceptor and TR-
FRET[44] were acquired (Fig. 1E, F and G). In agreement with the previous
results in transfected CHO-K1 cells, CXCR4 (Fig. 1E) and ACKR3 (Fig. 1F)
and ACKR3-ΔCterm (Fig. 1G) homo-oligomers were detected in
HEK293T cells. CXCR4 and its oligomers were mostly localized at the
membrane (Fig. 1E). In contrast to the specific signals obtained in Fig. 1E, no

specific d2 signal was detected upon treatment of mock-transfected
HEK293T cells with VUN400-d2 (Fig. S15A). Moreover, binding of
VUN400-d2 to Lumi4-Tb-labeled SNAP-CXCR4 was blocked by the
CXCR4 specific antagonist IT$_1$t (Fig. S15B). The membrane localization of
CXCR4 and its oligomers was in great contrast with the punctate signals
observed in ACKR3 expressing cells (Fig. 1F), that most probably reflects a
predominant localization of the receptor in intracellular compartments.
This suggests that the bound nanobodies can be internalized with the
receptor due to the constitutive internalization of ACKR3[45] (see Supple-
mentary Information, Figs. S15 and S16). The comparison of the labeling of
SNAP-ACKR3 and SNAP-ACKR3-ΔCterm which is devoided of any
internalization capacity[46] supported this hypothesis (see Supplementary
Information, Fig. S17). Indeed, as expected, in cells expressing the truncated
ACKR3-ΔCterm, more receptors were detected at the cell surface, as
compared to ACKR3-WT, reminiscent of CXCR4 localization (Fig. 1G).

In order to demonstrate the ability of the method to detect CXCR4 and
ACKR3 hetero-oligomers, HEK293T cells were co-transfected with the
constructs encoding Halo-CXCR4 and either SNAP-ACKR3 or SNAP-
ACKR3-ΔCterm. As shown on Fig. 2A and B, TR-FRET signals between the
nanobodies corresponding to the hetero-oligomers were detected in the cells
co-expressing Halo-CXCR4 and SNAP-ACKR3 or SNAP-ACKR3-
ΔCterm. A higher TR-FRET signal was observed at the cell surface in the
cells expressing SNAP-ACKR3-ΔCterm compared to SNAP-ACKR3-WT,
corroborating the higher cell-surface localization of the truncated receptor.
Moreover, in contrast to the cells co-expressing both receptors, only a very

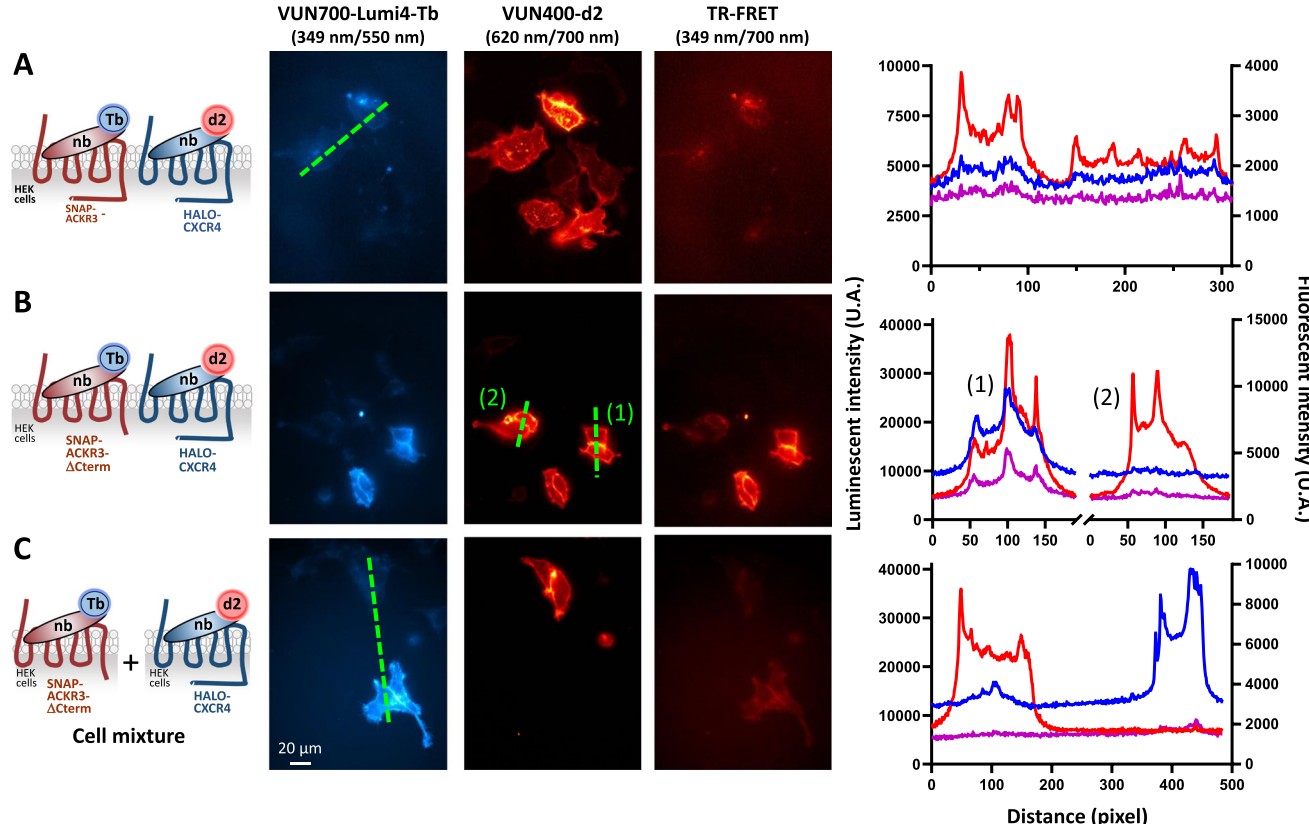

**Fig. 2 | Detection of CXCR4 and ACKR3 hetero-oligomers with labeled nano-bodies in TR-FRET microscopy.** HEK293T-cells were transfected with Halo-CXCR4 and ACKR3 (**A**), or Halo-CXCR4 and SNAP-ACKR3-ΔCterm (**B**) plasmid DNA. Transfected cells were incubated with VUN400-d2 and VUN700-Lumi4-Tb at a concentration of 10 nM for 2 h at 37°C. **C** TR-FRET signal was not due to collisional FRET. HEK293T cells transfected either with Halo-CXCR4 or SNAP-ACKR3-ΔCterm plasmid DNA were mixed together (mixed cells). Cells were then incubated with VUN400-Lumi4-Tb and VUN700-d2, and images of donor and acceptor fluorescence, as well as TR-FRET were acquired by the TR-FRET microscope, as described in Table S1. In all the conditions, quantification of the luminescence of the donor (blue line), the fluorescence of the acceptor (red line) and the TR-FRET signal (magenta line) was performed along the green lines in the images. No significant TR-FRET signal was recorded when expressing either HALO-CXCR4 or SNAP-ACKR3-ΔCterm (**C**). Scale bar: 20 μm.

low TR-FRET signal was detected in the cultures containing a mixture of cells expressing either CXCR4 or ACKR3-ΔCterm corresponding to the emission bleed-through of Lumi4-Tb at the FRET wavelength (Fig. 2C).

**Detection of endogenous CXCR4 oligomers in human cell lines**
We then analyzed the presence of endogenous CXCR4 homo-oligomers in non-transfected human cell lines. We first analyzed the CXCR4 and ACKR3 cell surface expression levels in CHO-K1, HEK293T and THP-1 cells (with no to low CXCR4 expression), and Raji, Jurkat and CCRF-CEM human leukemia/lymphoma-derived cell lines (endogenously expressing high levels of CXCR4) by flow cytometry (Fig. 3A). Because none of these cell lines had significant ACKR3 surface expression levels, our study was focused on CXCR4. Each cell line was incubated with a mixture of VUN400-Lumi4-Tb and VUN400-d2 (Fig. 3B). The signals of the donor, acceptor, and TR-FRET between the nanobodies were measured. Specific signals after the subtraction of the non-specific signal determined in the presence of 10 μM IT1t were plotted as a function of the nanobodies concentrations. In THP-1 and HEK293T cells that express low levels of CXCR4, we did not detect CXCR4 via binding of the labeled nanobody nor could we detect CXCR4 homo-oligomers by TR-FRET. The TR-FRET fits showed low R² values of 0.24 and 0.03 for THP-1 and HEK293T cells, respectively (Fig. 3C, D). For HEK293T cells, this was in agreement with the previous microscopy experiments, where we could not detect endogenous CXCR4 using the labeled nanobodies (Fig. S15A). In contrast, for the cell lines with higher CXCR4 expression levels, Raji, Jurkat, and CCRF-CEM, we detected CXCR4 expression with the signal from VUN400-Lumi4-Tb or VUN400-d2, as well as CXCR4 homo-oligomers with TR-FRET between the

nanobodies with high R² values for the TR-FRET fits of 0.97, 0.94, and 0.98 for Raji, Jurkat, and CCRF-CEM cells, respectively (Fig. 3E–G). TR-FRET$_{50}$ values for the TR-FRET signal determined in the cells were in the same range as the nanobody binding affinities (Table 1). Collectively, these results provide the first evidence for the existence of CXCR4 homo-oligomers in Raji, Jurkat and CCRF-CEM cell lines endogenously expressing CXCR4.

**Detection of endogenous CXCR4 oligomers in human PBMC**
The presence of CXCR4 oligomers in cancer cell lines can be due to receptor overexpression in pathological conditions. Therefore, we investigated the potential presence of endogenous CXCR4 homo-oligomers in physiological conditions, and specifically in PBMC. PBMC were incubated in the presence of VUN400- Lumi4-Tb (125 nM), VUN400-d2 (100 nM) or a mix of both nanobodies to investigate nanobody binding (signal of the donor or the acceptor) and receptor oligomerization (TR-FRET signal). At these concentrations (corresponding to 5xKd), 90% of CXCR4 is bound by a nanobody with an equal probability to bind either VUN400-Tb or VUN400-d2. Interestingly, a consistent FRET signal is observed in the presence of both nanobodies and is dramatically decreased in the presence of an excess of IT1t (Fig. 4). The effect of IT1t is due to a competition with the nanobodies for CXCR4 binding. These data strongly support the existence of oligomers in native PBMCs.

**Discussion**
The importance of GPCR oligomerization in physiology has been a matter of debate for more than three decades, as demonstrating the existence of receptor oligomers in native tissues remains highly

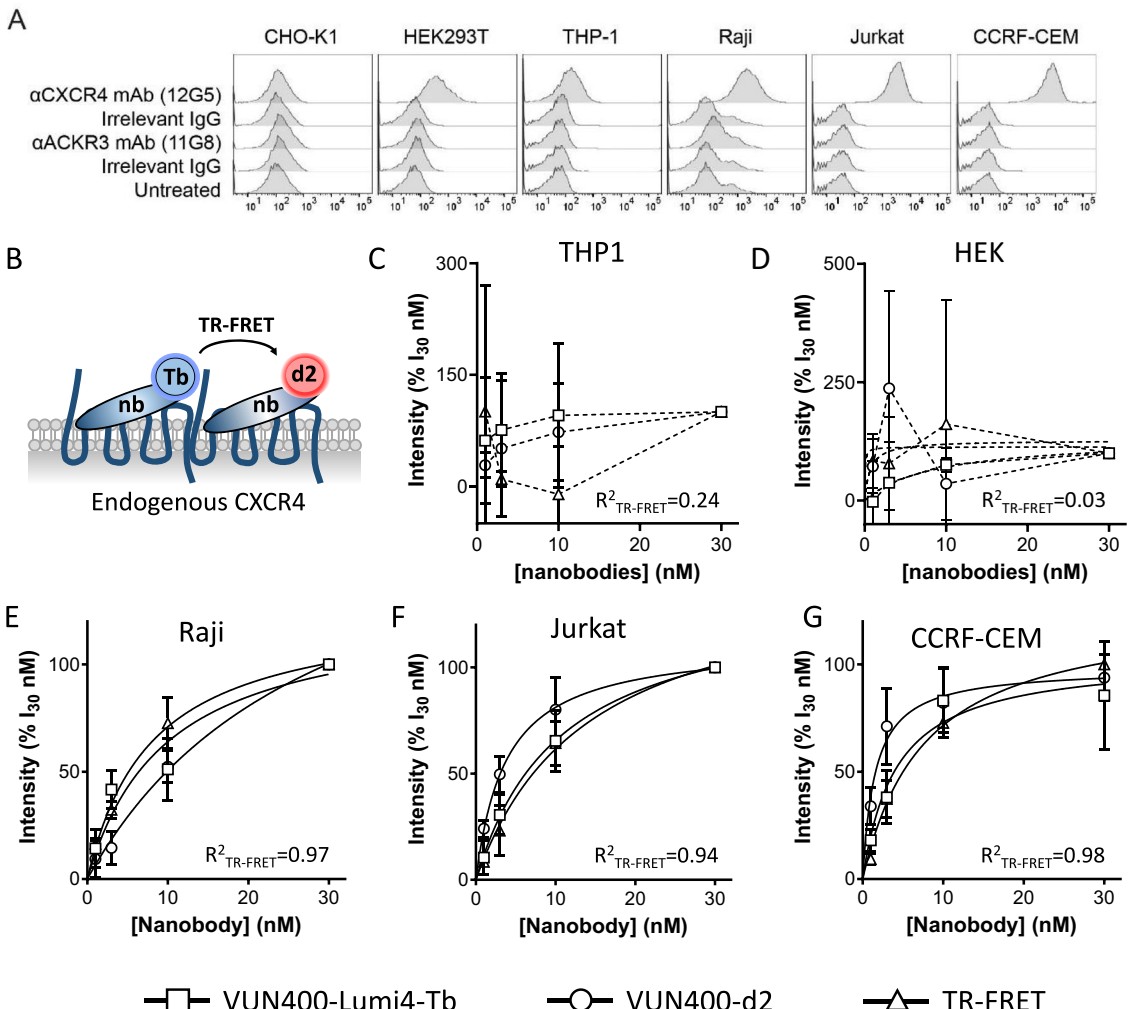

**Fig. 3 | Detection of endogenous CXCR4 homo-oligomers in human cancer cell lines. A** CXCR4 and ACKR3 surface expression levels. CHO-K1, HEK293T, THP-1, Raji, Jurkat and CCRF-CEM cells were evaluated by flow cytometry with antibodies against CXCR4 (12G5) and ACKR3 (11G8) and detected with secondary APC-labeled antibodies. In the same experiment, cells were treated with the corresponding isotype controls (Irrelevant IgG) or secondary antibodies alone (Untreated) as control conditions. **B** CXCR4 homo-oligomers were detected by determining TR-FRET signals upon incubation with a mixture of Lumi4-Tb- and d2-labeled VUN400 at equal concentrations. **C–G** Lumi4-Tb, d2 and TR-FRET signals obtained with a concentration range of labeled nanobodies in THP-1 (**C**), HEK293T (**D**), Raji (**E**), Jurkat (**F**), and CCRF-CEM (**G**) cells. Cells were incubated with different concentrations of labeled CXCR4 nanobodies. Lumi4-Tb-labeled nanobodies were excited at 337 nm and the emission wavelengths were detected at 620 nm (donor) or 665 nm (TR-FRET). The d2-labeled nanobodies were excited at 645 nm and their emission was captured at 680 nm. The specific binding signal was calculated by subtracting the non-specific binding signal determined in the presence of an excess of IT1t (10 µM) from the total binding signal. In addition, the contamination in the TR-FRET channel coming from the Lumi4-Tb donor has been subtracted. For each dataset, values were normalized to the signals obtained at with the highest concentration of nanobodies tested (30 nM). With the exception of those obtained in HEK293T cells ($n = 2$), means ± SDs of values obtained in three independent experiments are represented.

challenging. Various strategies have been used to study GPCR oligomers with RET-based techniques as some of the most powerful approaches because of the spatial resolution[47]. Most importantly, a RET signal can only be observed when a donor and acceptor are in close proximity, usually less than 8 nm. However, one of the major limitations is labeling receptors in a native context. The development of fluorescently-labeled nanobodies is a powerful strategy to achieve this goal: i) nanobodies can be generated to have a high affinity for GPCRs[48–51]; ii) they can be adapted via a fusion and conjugation with different molecules[52] and ligands[53]. In this latter case, the PTH-derived ligands have been fused to the C-term of the nanobodies using site-directed sortase-mediated labeling[53]; iii) finally the small size of the nanobodies ( ~ 15 kDa, 4 nm long and 2.5 nm wide) is advantageous to reduce the dynamic FRET that can be observed with fluorescently labeled larger conventional antibodies ( ~ 150 kDa)[34]. Therefore, we have developed fluorescently labeled nanobodies compatible with TR-FRET methods and directed them against the extracellular parts of CXCR4 and ACKR3. The labeled nanobodies demonstrated high binding affinities, as shown by measuring TR-FRET between the nanobodies and labeled receptors.

The TR-FRET technology between labeled nanobodies described in this study could be used for other receptors as well. Over the last decade, multiple nanobodies targeting GPCRs have been developed[54]. Although many of them target the intracellular parts of these receptors, several small antibody fragments targeting the extracellular epitopes of the chemokine receptors CXCR4[55–57], ACKR3[58], CXCR2[59], and the viral chemokine receptor US28[60] have also been described.

Using labeled nanobodies binding CXCR4 or ACKR3 we demonstrated the existence of CXCR4 and ACKR3 homo-oligomers and hetero-oligomers in transfected cells, both using 96-well plate format and TR-FRET microscopy, confirming previously reported findings[14,17–20,28]. No measurable or a weak expression of oligomers of ACKR3-WT were detected at the cell membrane, which points at the predominant intracellular localization of ACKR3 resulting from a constitutive internalization[61]. The detected intracellular TR-FRET with ACKR3-WT indicates that the nanobody probably

**Fig. 4 | Detection of endogenous CXCR4 homo-oligomers in peripheral blood monocyte cells (PBMC).** PBMC ($3.10^6$ to $6.10^6$ /point) were incubated in the presence of VUN400-Tb (125 nM) and/or VUN400-d2 (100 nM) with or without IT1t ($10^{-5}$M) at 4 °C for 2 h. Luminescence signals were acquired at different wavelengths: donor fluorescence: ex: 337 nm and em: 620 nm (left histogram); acceptor fluorescence: ex: 645 nm and em: 680 nm (middle histogram); TR-FRET signal: ex: 337 nm and em: 665 nm (right histogram). Signals were expressed as % of control: only donor for donor fluorescence, only acceptor for acceptor fluorescence, only donor for TR-FRET signal. ****: $p < 0.0001$.

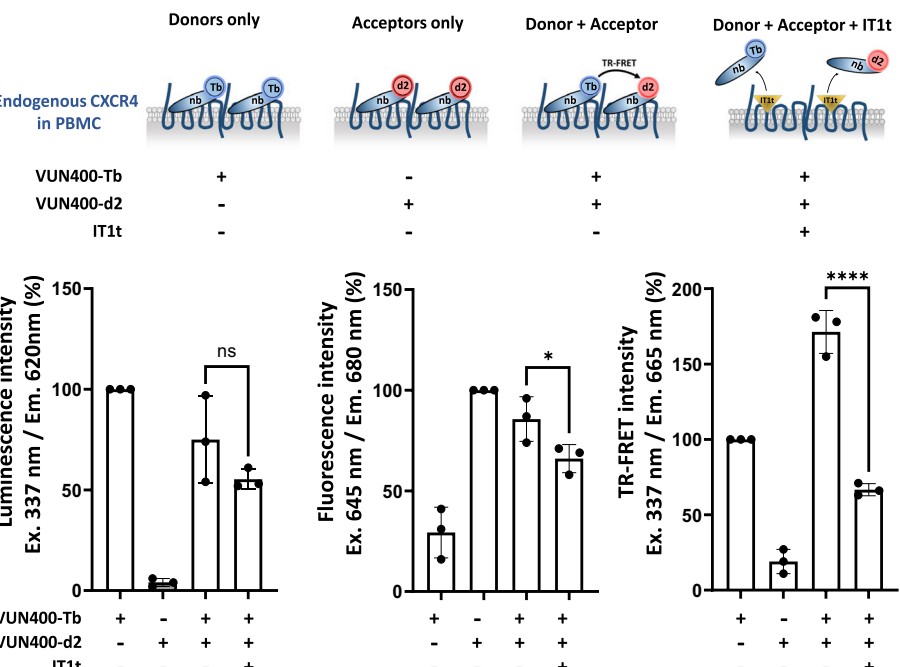

co-internalizes with the receptor. In contrast, a strong TR-FRET signal was observed at the plasma membrane of cells expressing a C-terminally truncated, non-internalizing, ACKR3-ΔCterm mutant.

Quantification of the percentage of oligomers is challenging when using TR-FRET. This is in part due to a large heterogeneity in the receptor complexes (dimer, tetramers and higher order oligomers), depending on the density of the receptor, and then variation from one cell to another. Moreover, receptor clusters can bind the differentially labeled nanobodies with varying ratio of 1/3, 2/2 or 3/1, giving rise to multiple resonance energy transfer possibilities. These factors affect the distances between the donor and acceptor molecules, which significantly impacts TR-FRET signals due to its sixth power dependence on the distance between donor and acceptor. Hence, when the population is not homogeneous, TR-FRET signals do not correlate to number of oligomers. Furthermore, TR-FRET microscopy does not make use of a confocal plane and therefore does not allow for specific investigation of internalized oligomers compared with those remaining on the cell surface.

With the described nanobody-based TR-FRET approach, we detected for the first time endogenous CXCR4 homo-oligomers in lymphoma- or leukemia-derived human cell lines and in PBMCs. Multiple cancer cells show elevated expression of CXCR4, including leukemia, T- and B cell lymphoma cells[62]. CXCR4 overexpression is associated with poor clinical outcome[63,64]. By contrast, the expression of CXCR4 in PBMCs is highly heterogeneous ranging from few hundreds to more than 50,000 receptor per cell[65]. This can lead to a similar heterogeneity of CXCR4 oligomer density in PBMCs since receptor oligomerization is dependent on receptor density[21]. Hence, the higher expression level might result in an increase in CXCR4 oligomers. The detection of CXCR4 oligomers in the cancer cells suggests that they might contribute to cancer development and/or progression. Indeed, CXCR4 oligomers have been demonstrated to be involved in cell migration[18,20] and G-protein signaling[19]. However, these oligomers might also be present in non-cancerous cell lines and play a role not only in pathology. Hypothetically, as shown in Martinez-Munoz et al. [20], oligomer formation might be at the cost of the amount of CXCR4 monomers and suggests that monomers and oligomers have distinct signaling properties. The observed TR-FRET between labeled nanobodies provides compelling evidence on the existence of endogenous CXCR4 oligomers in both cancerous and physiological contexts.

We could not detect CXCR4 oligomers using our approach in the cells expressing low CXCR4 densities. This might suggest that oligomerization only occurs when CXCR4 expression reaches a minimal level, as previously reported for other GPCRs[66]. Alternatively, the sensitivity of our TR-FRET approach might be insufficient to detect low expression or low oligomer amounts. For example, in HEK293T cells CXCR4 can be detected via flow cytometry, but not with the labeled nanobodies in 96-well plates or TR-FRET-based microscopy. In addition, a proportion of homo-oligomers in these experiments is not detected by the TR-FRET signal when two donor or two acceptor molecules are bound to one dimer, or if a negative binding cooperativity exists between two nanobodies.

In conclusion, we have successfully applied nanobody labeling combined with TR-FRET to reveal the presence of endogenous CXCR4 oligomers opening new avenues to study its oligomerization in a disease context, e.g., in cancer. This approach can be adapted to other GPCRs and membrane-bound proteins to study their supramolecular organization under pathophysiological-relevant conditions.

## Materials and methods
### Protocols for cell culture, cell transfection, nanobody selection and fluorescence labeling are described in details in supplementary information
**Cell culture and transfection.** THP-1 (ATCC® TIB-202™, human monocytes, acute monocytic leukemia), Raji (ATCC® CCL-86™, human B-lymphocytes, Burkitt's lymphoma), CCRF-CEM (ATCC® CCL-119™, human T-lymphoblasts, acute lymphoblastic leukemia), Jurkat (ATCC® TIB-152™, human T-lymphocytes, acute T-cell leukemia), CHO-K1 (ATCC® CCL-61™, Chinese Hamster ovarian), and HEK293T cells (ATCC® CCRL-3216™, human embryonic kidney) were purchased from ATCC. THP-1, CCRF-CEM and Jurkat cells were cultivated in suspension in Roswell Park Memorial Institute (RPMI) 1640 medium containing 10% fetal bovine serum (Thermo Fisher Scientific, Waltham, MA, USA) and 1% Penicillin/streptomycin. HEK293T cells were cultivated in Dulbecco's Modified Eagle Medium containing 10% fetal calf serum and 1% Penicillin/streptomycin. CHO-K1 cells were cultured in the same medium, but 1% non-essential amino acids (Thermo Fisher Scientific) were added.

CHO-K1 were transfected using JetPEI® according to the manufacturer's protocol (Polyplus Transfection). CHO-K1 cells were seeded

one day before the transfection at a density (cell/well) of 30,000 in 96 well plate, 300,000 in a 6-well plate and 1,500,000 in 150 mm dishes. Quantity of plasmids was adapted according to the manufacturer recommendation.

HEK293T cells were transfected with Lipofectamine2000 (Invitrogen, Thermo Scientific) according to the manufacturer protocol. For TR-FRET microscopy, cells were seeded at a density of 50,000 in four-well chambered coverglasses (ThermoFisher Scientific) coated with poly-ornithine.

**Development of ACKR3-specifc nanobody via phage-display.** Two llamas were immunized with ACKR3-encoding plasmid DNA as described previously[67]. Nanobody phage-display libraries were constructed according to the previously described method[68]. Briefly, selections for ACKR3-specific binders were performed using phage panning on ACKR3-expressing or empty (null) virus-like particles (Integral Molecular, Philadelphia, PA, USA) immobilized in MaxiSorp plates (Nunc, Roskilde, Denmark). Three consecutive rounds of selection were performed. The isolated ACKR3-binding nanobodies fused with His6 tags were purified onto Nickel Excel Sepharose™ column (GE Healthcare, Chicago, IL, USA) and eluted with a buffer containing 500 mM imidazole (Sigma-Aldrich, St. Louis, MO, USA). The elution buffer was exchanged for Dulbecco's phosphate buffered saline (DPBS; Thermo Fisher Scientific) by desalting using Amicon® Ultra-0.5 Centrifugal Filters with molecular weight cut-off of 3000 (Sigma-Aldrich). Amino acid sequences of the selected ACKR3 binding nanobody VUN700, as well as the CXCR4 binding nanobody VUN400, are shown in Table S1.

**Sortase A pentamutant production.** The sortase enzyme from *Staphylococcus aureus* has a transpeptidase activity and catalyzes the covalent binding of LPXTG amino acid moiety to polyglycine peptides resulting in a LPXTGGG sequence[69]. To maximize the yield of the nanobody labeling, we used the pentamutant version of sortase with increased LPETG-coupling activity[70]. The sortase A pentamutant plasmid (Addgene, MA, USA) was transformed in BL21 competent *E. coli* bacteria. One colony was transferred into 100 mL liquid Lysogeny broth (LB) containing 0.1% Kanamycin and was grown overnight at 37 °C while agitating. 50 mL of the preculture was grown in 2 L of liquid LB containing 0.1% Kanamycin in flasks at 37 °C while agitating. When an OD600 of approximately 0.7 was reached, 2 mL of 0.5 mM IPTG was added and bacteria were further incubated at 20 °C overnight while agitating. The bacterial culture was centrifuged at 6000 x *g* for 30 min at 4 °C and cell pellets were kept at −80 °C for 1 h. The frozen pellets were resuspended at room temperature in 20 mL of lysis buffer (50 mM Tris pH 7.4, 150 mM NaCl, 100 μM phenylmethanesulfonil fluoride, 10 μg/mL lysozyme, 10 μg/mL benzamidine, and 10 μg/mL leupeptin (Sigma-Aldrich). The lysed bacteria were sonicated for 3 × 2 min alternating 1 s sonication with 1 s pauses. Then they were centrifuged at 15,000 g for 15 min at 4°C. The lysate was collected and imidazole was added obtaining a final concentration of 15 mM. The His-tagged sortase protein was purified on a Ni-NTA column. The lysate was loaded on the Ni-NTA column and was washed with 10 CV of washing buffer containing 50 mM Tris pH 7.4, 150 mM NaCl and 30 mM imidazole. The sortase protein was eluted with an elution buffer consisting of 50 mM Tris pH7.4, 150 mM NaCl and 500 mM imidazole. The fractions containing protein (tested with Bradford) were pooled together and concentrated using a 15 kDa cellulose membrane. The sortase protein was isolated using size exclusion chromatography (SEC) in a buffer containing 50 mM Tris pH 7.4 and 150 mM NaCl and then concentrated again on a 15 kDa cellulose membrane. The sortase A pentamutant protein was stored at −80 °C containing 50% glycerol.

**Synthesis and preparation of peptides for labeling.** $H_2N$-GGGC-COOH and $H_2N$-CDYKDDDDLPETGG-COOH peptides were synthesized by solid-phase peptide synthesis at the platform of Synbio3 (Montpellier, France). The peptides (1 eq., 0.1 mmol) were synthesized

on Wang resin using HATU (20 eq., 2 mmol) and DIPEA (40 eq., 4 mmol) in DMF for amide coupling, and 20% piperidine in DMF for deprotection using the Liberty Blue™ Peptide Synthesizer (BioSPX) without microwave assistance. Each amino acid (4 eq., 0.4 mmol) was coupled twice for 30 min at room temperature. After the final amino acid coupling and deprotection, peptides were cleaved from the resin and side-chain deprotected with a solution of TFA/TIPS: 94/6 (10 mL for 4 h). The resin was washed with fresh TFA (5 mL) and the solution concentrated in vacuo. The resulting oils were precipitated with ice-cold ether and the supernatant was removed. The precipitate was solubilized in water and lyophilized. Crude peptides were suspended in a minimum volume of a solution of $H_2O/CH_3CN$: 80/20. Purification was performed using HPLC with a preparative column on a gradient of $CH_3CN/H_2O$ (with 0.1% formic acid) for 45 min. The following gradients were used for purifying:

– $H_2N$-CDYKDDDDLPETGG-COOH: 0% of $CH_3CN$ for 5 min, 0–5% of $CH_3CN$ for 5 min, 5–15% of $CH_3CN$ for 20 min, 15–100% of $CH_3CN$ for 3 min and 100% of $CH_3CN$ for 5 min.
– $H_2N$-GGGC-COOH: 0% of $CH_3CN$ for 5 min, 0–20% of $CH_3CN$ for 10 min, 20–100% of $CH_3CN$ for 1 min and 100% of $CH_3CN$ for 14 min.

Fractions were checked by LCMS, concentrated in vacuo, lyophilized and stored at 4 °C.

The cysteines of the peptides were labeled by Cisbio using thiol-reactive Tb and d2 fluorophores.

**Fluorescent labeling of peptides**
GGGC-d2. To a solution of peptide GGGC (9 mg, 30.8 μmol) in dry DMSO (200 μL) was added a solution of d2-maleimide (2.5 mg, 3.1 μmol) in Hepes buffer (100 mM, pH 6.5, 3 mL). The resulting mixture was stirred at room temperature for 1 h and then purified by preparative HPLC (gradient 1) leading to the title compound as a pure blue powder (1.9 μmol, 61%). UPLC (gradient 2), Rt 1.94 min. LRMS (ESI+) : $[M + H]^+$, $m/z = 1057$.

d2-CDYKDDDDKLPETGG. To a solution of d2-maleimide (2.5 mg, 3.1 μmol) in Hepes buffer (100 mM, pH 6.5, 3 mL) was added peptide CDYKDDDDKLPETGG (5.2 mg, 3.1 μmol). The resulting mixture was stirred at room temperature for 1 h and then purified by preparative HPLC (gradient 1) leading to the title compound as a pure blue powder (1.4 μmol, 45%). UPLC (gradient 2), Rt 1.99 min. LRMS (ESI+) : $[M + 2H]^{2+}$, $m/z = 1219$.

GGGC-Lumi4-Tb. To a solution of peptide GGGC (9 mg, 30.8 μmol) in dry DMSO (200 μL) was added a solution of Lumi4-Tb-maleimide (4.82 mg, 3.0 μmol) in Hepes buffer (100 mM, pH 6.5, 3 mL). The resulting mixture was stirred at room temperature for 1 h and then purified by preparative HPLC (gradient 3) leading to the title compound as a pure white powder (1.2 μmol, 40%). LCMS (gradient 4), Rt 9.25 min. LRMS (ESI+) : $[M + H]^+$, $m/z = 1721$.

Lumi4-Tb-CDYKDDDDKLPETGG. To a solution of Lumi4-Tb maleimide (4.8 mg, 3.0 μmol) in Hepes buffer (100 mM, pH 6.5, 3 mL) was added peptide CDYKDDDDKLPETGG (6.5 mg, 3.9 μmol). The resulting mixture was stirred at room temperature for 1 h and then purified by preparative HPLC (gradient 3) leading to the title compound as a diastereomeric mixture white powder (2.2 μmol, 73%). UPLC (gradient 4), Rt 11.94-12.46 min. LRMS (ESI+) : $[M + 2H]^{2+}$, $m/z = 1550$.

Analytical HPLC have been run with ThermoScientific P4000 connected to UV 1000 detector (deuterium lamp, 190–350 nm). Preparative HPLC have been run with Shimadzu LC-8A connected to a diode-array detection Varian ProStar. UPLC-MS have been run with a Waters Acquity Hclass connected with detector simple quadripole SQD2 in positive mode electrospray (capillary 3.2 kV, cone tension 30 V). Waters Xbridge $C_{18}$, 5 μm, 19 × 100 mm column is used for gradient 1 and gradient 3 with 20 mL.min$^{-1}$ flow rate. Waters Acquity $C_{18}$, 300 Å, 1.7 μm, 2.1 × 50 mm column is used for gradient 3 with a flow rate of 0.6 mL.min$^{-1}$. Waters

Acquity C$_{18}$, 300 Å, 3.5 μm, 4.6 × 100 mm column is used for gradient 4 with a flow rate of 1 mL.min$^{-1}$.

Gradient 1: Solvent A: H$_2$O-0.2% TFA; Solvent B: MeCN; Gradient: $t = 0$ : 10%, B − $t = 19$ min : 50% B. Gradient 2: Solvent A: H$_2$O-0.1% HCO$_2$H; Solvent B: MeCN 0.1% HCO$_2$H; Gradient: $t = 0$ : 10% B − $t = 0.2$ min : 10% B − $t = 4.9$ min : 60% B. Gradient 3: Solvent A: H$_2$O-25 mM TEAAc pH7; Solvent B: MeCN; Gradient: $t = 0$ : 5%, B − $t = 19$ min : 40% B. Gradient 4: Solvent A: H$_2$O 25 mM TEAAc pH7; Solvent B: MeCN; Gradient: $t = 0$ : 1% B − $t = 19$ min : 20% B.

**N-terminal sortase-mediated labeling of nanobodies with the peptides conjugated with the fluorescent probes.** Nanobodies were modified at their N-terminus with a GGG sequence. 50 μM nanobodies, 250 μM Tb-CDYKDDDDKLPETGG or d2-CDYKDDDDKLPETGG peptides, and 2.5 μM sortase A pentamutant were incubated in the sortase buffer (50 mM Tris pH 7.4, 150 mM NaCl, 10 mM CaCl$_2$) on ice for 2 h 45 min. The labeled peptides contained FLAG tag (DYKDDDDK) in their sequence. After the incubation, the mixture was loaded on a SEC column (Superdex 75, 10/300) and fractions of the peak corresponding to the labeled nanobody was loaded on a Flag M2 column. The labeled nanobody was eluted from the column with an excess of Flag peptides and the sample was again injected on a SEC column (Superdex 75, 10/300). The collected fractions containing the labeled nanobody were concentrated on a 10 kDa cellulose membrane, aliquoted and stored at −80 °C.

**C-terminal sortase-mediated labeling of the nanobody with the peptides conjugated with the fluorescent probes.** Nanobodies were modified at their C-terminus with an LPETG sequence followed by a His6 tag. 50 μM nanobodies, 250 μM GGGC-Tb or GGGC-d2, and 2.5 μM sortase A pentamutant containing a His6 tag were incubated in the sortase buffer (50 mM Tris pH 7.4, 150 mM NaCl, 10 mM CaCl$_2$) on ice for 2 h 45 min. During this incubation, sortase catalyzes the fusion of LPETG sequence with the GGC-Tb or GGC-d2 fluorescent peptides, releasing the His-tag previously linked to the nanobody. The mixture was then loaded on a Ni-NTA column that was pre-washed with sortase buffer containing 30 mM imidazole. Sortase containing a His-tag was retained on this column. The collected fractions from the flow through containing the labeled nanobody, verified by a Bradford assay, were concentrated on a 10 kDa cellulose membrane and then injected on a SEC column (Superdex 75, 10/300). The purified labeled nanobody fractions corresponding to a peak in the SEC profile were again concentrated, and their concentration was measured by measuring absorbance at 280 nm wavelength using a DU720 General Purpose UV/VIS spectrometer (Beckman Coulter). Nanobodies were stored at −80 °C. The number of d2 molecules per nanobody was calculated by dividing the amount of fluorophores (650 nm/ε(d2)) by the amount of nanobody (280 nm/ε(nanobody)), where ε is the extinction coefficient. In the similar way, the number of Lumi4-Tb molecules per nanobody was calculated by dividing the amount of fluorophores (340 nm/ε(Lumi4-Tb)) by the amount of nanobody (280 nm/ε(nanobody)). The number of fluorophores per nanobody was between 0.6 and 0.9.

**Acquisition of the fluorescent signals.** All the luminescent and fluorescent signals in the time-resolved Förster resonance energy transfer (TR-FRET) experiments on cells in plates have been acquired on a Pherastar plate reader (BMG Labtech). For the specific wavelengths, refer to Table S1.

**Bioluminescence resonance energy transfer (BRET) experiments.** On day 1, 300,000 CHO-K1 cells were seeded per well in 6-well plates. 24 h later, cells were transfected using JetPEI ® with pcDNA3-CXCR4-nLuc (600 ng per well), pcDNA3-CXCR4-YFP (300 ng per well), and an empty vector (PRK5, 3100 ng per well) to obtain a total DNA concentration of 4 μg per well, according to the manufacturer's protocol

(Polyplus Transfection, NY, USA). On day 3, transfected cells were harvested and seeded into a white 96-well plate at a density of 30,000 cells per well. On day 4, the plate was washed twice with 100 μL of Phosphate Buffered Saline (PBS) and then the BRET signal was measured by adding 5 μM coelenterazine. Nanobodies were added at different concentrations and dose-response curves were generated using Graphpad Prism 6®. mBRET (mBRET = {(535/480)$_{assay}$ − (535/480)$_{nLuc alone}$} × 1000) was plotted as a function of ligand concentration. The pcDNA3-CXCR4-nLuc plasmid was a kind gift from Ali Işbilir (Max Delbrück Center for Molecular Medicine, Berlin, Germany). Two-way ANOVA was used to determine the statistical significance of each effect. A Tukey correction factor was implemented to correct for errors coming from multiple comparisons.

**TR-FRET cell binding assay with labeled nanobodies.** The day after the transfection, CHO-K1 cells were transferred in 96-well plate at a density of 30,000 cell/well. On day 2 after the transfection, Tag-lite-based saturation or competition experiments were performed[31]. Briefly, cells were incubated at 37°C for 1 h with Halo- or SNAP-Lumi4-Tb (100 nM) or Halo- or SNAP-red (300 nM) substrates (Cisbio Bioassays). After washing steps, cells were incubated with different concentrations of fluorescent nanobodies and in the presence (non-specific binding) or the absence (total binding) of an excess of receptor-specific competitors: a CXCR4 antagonist 10 μM IT1t (R&D Systems), or 300 nM VUN700 (unlabeled Nb) and 31.6 nM CXCL12 (NS, R&D Systems). Luminescent signals of donor and TR-FRET were acquired (see Table S1). The specific binding was plotted as a function of nanobody concentration. Curves were analyzed with GraphPad Prism 6®. In the saturation binding experiments, curves were fitted with the "one site-specific binding curve" subroutine to determine the affinities of the nanobodies.

**TR-FRET microscopy with labeled nanobodies.** Experiments were performed on CXCR4- and/or ACKR3-transfected HEK293 cells in four-well chambered coverglasses (ThermoFisher Scientific) (see Supplementary Information). Lumi4-Tb- and d2-labeled nanobodies were added to the chambers at a concentration of 10 nM and incubated for 4 h at 37 °C. Then, chambers were washed and images were immediately acquired using our in-house TR-FRET microscope as described previously[44]. For the conditions of signal acquisition see Table S1. Images were treated with ImageJ®.

**Detection of CXCR4 and ACKR3 homo-oligomers in transfected CHO-K1 cells.** One day after the transfection, CHO-K1 cells seeded in a 150 mm plate (see Supplementary Information) were harvested and aliquots of 500,000 cells were incubated for 2 h at room temperature with various concentrations of nanobodies. Nonspecific labeling was performed in the presence of 10 μM IT1t and/or 10 μM VUF15485, a specific competitor of CXCL12 for ACKR3 and exhibiting agonist properties (kindly provided by Rob Leurs, Griffin Discoveries, Amsterdam), 300 nM unlabeled VUN700 and 31.6 nM CXCL12. After a quick wash step, donor, acceptor and TR-FRET signals were acquired (see Supplementary Information). Fluorescence signals were plotted as a function of nanobody concentrations and analyzed with GraphPad Prism 6®. Curves were fitted with the "one site-specific binding curve" subroutine.

TR-FRET signals resulting from collision between transfected CHO-K1 cells (1 million cells/sample) were determined on a mix (1:1) of cells transfected either with HALO-CXCR4 plasmid or SNAP- ACKR3-ΔCter plasmid.

**Determination of the detection limit for CXCR4 homo-oligomers in transfected CHO-K1 cells.** TR-FRET signal resulting from receptor oligomerization was plotted as a function of receptor expression. CHO-K1 cells plated in 6-well plates were transfected using JetPEI® (see Supplementary Information) with increasing amounts of pcDNA3-Halo-CXCR4. One day after transfection, cells were harvested and seeded into

**Article**

two white 96-well plates (30,000 cells/well). Two days after transfection, receptor expression was estimated by the amplitude of the luminescent signal of Halo-Lumi4-Tb after incubation of the cells with 100 nM Lumi4-Tb Halo substrate (Table S1). The TR-FRET signal after the incubation of cells in the presence of 20 nM VUN400-Lumi4-Tb alone or 20 nM VUN400-Lumi4-Tb with 20 nM VUN400-d2 was measured in the presence or the absence of 10 μM IT1t. The non-specific TR-FRET signal (assessed in the presence of IT1t) was subtracted from the total TR-FRET. Values were normalized to the CXCR4 expression levels and curves were fitted with GraphPad Prism 6®.

**Analysis of CXCR4 and ACKR3 cell surface expression by flow cytometry.** CHO-K1, HEK293T, THP-1, Raji, Jurkat and CCRF-CEM cells were incubated with anti-CXCR4 (clone 12G5, R&D Systems, Minneapolis, MN, USA) or anti-ACKR3 antibodies (clone 11G8, R&D Systems, Minneapolis, MN, USA), or corresponding isotype control antibodies for 60 min at +4 °C in PBS containing 0.5% FBS. THP-1 cells were pre-incubated for 30 min with human BD Fc block reagent (BD Biosciences, Franklin Lakes, NJ, USA) before adding the antibodies. Fc block reagent was used during all incubation steps with THP-1 cells. After the incubation with primary antibodies, the cells were washed three times in PBS containing 0.5% FBS and incubated with secondary anti-mouse polyclonal Ig Multiple Adsorption APC-conjugated (polyclonal, BD Biosciences). Cells were washed and resuspended in PBS containing 0.5% FBS and analyzed using a FACSCanto™ flow cytometer (BD Bioscience). FlowJo® 10 software (FlowJo, LLC, Ashland, OR, USA) was used for data analysis.

**Detection of endogenous CXCR4 oligomers in cancer cell lines.** THP-1, Raji, CCRF-CEM and Jurkat cells were centrifuged at 800 g for 10 min. and resuspended in RPMI-1640 medium at a density of 10 million cells/mL. 100 μL aliquots of this cell solution have been incubated with different concentrations of the labeled nanobodies for 2 h at room temperature in the presence or absence of 10 μM IT1t. Medium with the nanobodies was removed, cells were resuspended in 100 μL of TagLite® buffer, transferred to 96-well plates, and kept on ice. The Lumi4-Tb, d2 and TR-FRET signal were measured at 620, 680 and 665 nm, respectively using the Pherastar plate reader. The non-specific binding signal, obtained by incubating with IT1t, was subtracted from the total binding signal. Emission ratio (665/620) was plotted as a function of the nanobody concentrations and data analyzed with GraphPad Prism 6®. Curves were fitted with the "one site-specific binding curve" subroutine.

**Detection of endogenous CXCR4 oligomers on peripheral blood mononuclear cells.** Purified PBMCs were kindly provided by Cisbio. PBMCs were resuspended in tag-lite buffer at concentrations ranging from 50.10⁶ to 80.10⁶ cells/ml. VUN400-Lumi4-Tb and VUN400-d2 were used at a final concentration of 125 nM and 100 nM, respectively. PBMCs (3 to 6 million of PBMC/point) were incubated 4 h at 4°C, centrifuged and washed two times with Taglite buffer maintained at 4 °C, and suspended in a final volume of 60 μL. A volume of 50 μl were transferred in a white 96-well plate. Luminescent signals of the donor (ex: 337 nm; em: 620 nm) of the acceptor (ex: 640 nm; em: 680 nm) and of the TR-FRET signal (ex: 337 nm; em: 665 nm) were measured.

**Statistical analysis**
Statistics were performed using a one-way ANOVA analysis with a Šídák's (Fig. S14) or Tukey's (Fig. 4) multiple comparisons test *: $p < 0.1$; **: $p < 0.01$; ***: $p < 0.001$; ****: $p < 0.0001$.

**Reporting summary**
Further information on research design is available in the Nature Portfolio Reporting Summary linked to this article.

## Data availability

The authors declare that the data supporting the findings of this study are available within the paper and its Supplementary Data. Should any raw data files be needed in another format they are available from the corresponding author upon reasonable request.

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

## Acknowledgements

This work was supported by the European Union's Horizon 2020 MSCA Program under grant agreement 641833 (ONCORNET) (J.H., V.B., A.F., F.B., P.M., M.J.S., T.D.). This work was also supported by CNRS, INSERM, University of Montpellier, the Fondation de la Recherche Médicale (J.H.—Grant Number 184561) and Agence Nationale de la Recherche (grant agreement 194762), CMG has benefited from the support by the Labex EpiGenMed, an "Investissements d'avenir" program, reference ANR-10-LABX-12-01. Thanks to Dr. Pascal Verdié and the platform Synbio3 (Faculty of Pharmacy, Montpellier, France) for the peptide syntheses. This work was also made possible thanks to the facilities of Biocampus platform Arpege of Montpellier and Languedoc-Roussillon. Thanks are due to Dr P. Rondard for giving us the fluorescent nanobody labeled with d2 (DN1-d2) targeting mGluR2 and to A. Aquilina from Perkin Elmer—Cisbio for providing us PBMC.

## Author contributions

J.H., V.B., T.D. and M.J.S. designed the study; V.B., B.v.d.W. and H.d.H. generated and characterized the nanobodies; J.H., C.M.G., L.L., E.T., F.B., T.D. constructed the labeled receptors and nanobodies; C.B. analyzed the labeled nanobodies; J.H., V.B., C.M.G., M.L., O.O. and T.D. performed binding experiments and TR-FRET microscopy. J.H., V.B., C.M.G., A.F., C.B., L.L., E.T., F.B., P.M., B.M., S.G., B.v.d.W., H.d.H., R.H., M.J.S. and T.D. wrote and/or reviewed the manuscript.

## Competing interests

The authors declare no competing interests.
