## [Transparent Peer Review file · Communications Biology]

TR-FRET between engineered nanobodies reveals the existence of endogenous CXCR4 oligomers

Corresponding Author: Dr Thierry Durroux

Version 0:

Reviewer comments:

Reviewer #1

(Remarks to the Author)

This a novel study describing the use of nanobodies to investigate GPCR oligomers. Overall, the study is very interesting, and makes a great contribution to the field. The paper itself requires a little more work till it is publication ready.

Review: general comments

There is a Table S1 and a Table S2 but no Table S3?

The authors should conduct some experiments to confirm the selectivity of the nanobodies. Does VUN-400 bind to ACKR3 and does VUN-700 bind to CXCR4?

Can the authors please explain how three unlabelled ligands (unlabelled VUN700, CXCL12, and VUF15485 – from line 629 and in Fig S11) have been used to generate the specific binding curve shown in Fig 1B? Usually only a one unlabelled ligand would be used to generate specific binding, so this doesn't make sense.

Fig 1 legend – some of the wording for B is in A, i.e. talking about ACKR3 in A, but that should be in B.

Table 1 does not appear to have any EC50 values, only Kd values, even though Line 155 and the table legend mentions EC50s. I think the authors mean to say Kd Values?

Line 162 – “The maximum corresponding” would be clearer as “The maximum signal corresponding”

Line 172 – “Because random collisions occurrence varies as a power two of the receptor density, plotting TR-FRET as a function of receptor density would have led to a parabolic curve⁴²” I think this needs more clarification, I can't see in the referenced article where it says a linear curve indicates specific interactions while parabolic curve indicates random collisions.

Figure S12 – this is a very confusing figure. The figure legend needs to be reworded substantially. Specifically, it would be better to move this description of the graphs to the beginning of the legend:

“Transfected CHO cells similarly expressing HALO-CXCR4 and SNAP-mGluR2 receptors (B) are incubated in the presence of a mix of VUN400-Tb, VUN400-d2 and DN1-d2 nanobodies⁴, the latter targeting mGluR2 receptors (A). Donor (C), acceptor (D) and TR-FRET (E) luminescence signals are recorded. Despite significant labeling with VUN400-Tb (C) and/or DN1-d2 (D), no significant specific TR-FRET signal is observable between the nanobodies (E). By contrast, while the labeling with VUN400-d2 is much less intense than that observed with DN1-d2 (D), specific TR-FRET is observed between VUN400-Tb and VUN400-d2 (E). These data strongly suggest that TR-FRET signal is due to the existence of specific interactions between CXCR4 receptors and not to random collisions between receptors.”

By having it at the beginning of the figure legend, the reader immediately understands what the figure is about. Currently, less important details about the figure come first, which makes the legend confusing to read. Also, I assume that in panel A, the blue receptors are CXCR4 and the green ones are mGluR2, but again, this isn't specified in the legend, which it should be. Panel A also needs more explanation.

Figure 2D – the graph in this panel needs more explanation. How are the specific binding signal and the total binding signal determined in this graph? It is not stated in the figure legend.

Figure 3 – I think it would be good to compare the overall FRET values in these graphs. I assume C and D would have lower FRET values than E, F, G, but because all the data is normalised, that comparison can't be made. Also, as in Fig 2, how are the specific binding signal and the total binding signal determined in this graph? It is not stated in the figure legend. Also, Fig 3 D is only n=2. Presumably this should be n=3.

Review: specific comments

Line 47 – “Nevertheless, oligomerization of GPCRs was mostly investigated” should be “Nevertheless, oligomerization of GPCRs has mostly been investigated”

Line 50 – “previously referred to as CXCR7” - I would put this phrase in brackets rather than commas, as it currently reads a bit ambiguously whether “previously referred to as CXCR7” refers to just ACKR3 or both CXCR4 and ACKR3.

Line 134 – “resulted” should be “resultant”.

Figure S12B – one of the error bars appears to be white so is not visible. It should be changed to black like all the other error bars.

Line 248 – “CXCR4 oligomers in cancer cell line” should be “CXCR4 oligomers in cancer cell lines”

Line 250 – “and especially” – “specifically” would be more appropriate wording here.

Line 277 – “directed against” should be “directed them against”

Line 297 – “vary” should be “variation”

Supp Methods:

Second paragraph, second line – “CHO-K1 cells. Cells were” should be “CHO-K1 cells were”.

Fourth paragraph, third line – “method 7 Briefly” should be “method 7. Briefly”.

Fifth paragraph, third line – “refer to TableS2” should be “refer to Table S1”.

Reviewer #2

(Remarks to the Author)

This manuscript describes the development of CXCR4 and ACKR3 directed nanobodies, their labelling and assessment of the ability of these nanobodies to detect receptor oligomerization, with the intent to be able to detect this in natively expressing systems. This is a very good idea because, as the authors point out, dimerization of GPCRs remains controversial and there are no reliable data (that I'm aware of) to support dimerization in native settings. Unfortunately, this is let down by the poor design of experiments, lack of controls, inappropriate analysis and over-interpretation. As a consequence, there are no conclusions that are drawn that can be adequately supported by the presented data set.

The specificity of the nanobodies used is only validated in CHO cells through competition using CXCL12. These nanobodies may still recognise other receptors/proteins (in particular other chemokine receptors) that are not expressed on these cells. The contention of the manuscript is that the nanobodies will be able to be used to detect oligomerization in native tissues. Minimally this needs to be supported by data that demonstrates these are not cross reactive with closely related chemokine receptors. Ideally, binding to PBMC where their relevant targets have been knocked out would make sense.

Figure 1/table 1.

The apparent affinity of all antibodies in this data set are almost a full log lower than in the taglite assay – this needs to be addressed. In the data fit, the data are expressed as %max, yet the max will clearly be over 100%, which makes no sense. In the table affinities are reported with linear error, data is reported as being fitted using a 1-site binding model in PRISM, this model assumes mass-action and therefore error should be log normally distributed. The data does not make any sense, whatsoever. The TR-FRET signal tracks perfectly with fluorescence intensity from the individual antibodies. At concentrations below full occupancy, and in particular at concentrations below 50% occupancy, one would expect a mixture of singly occupied dimers and doubly occupied dimers. This will result in a sigmoidal TR-FRET signal. The only ways I can envision getting the result that is reported is either the filters used are not actually separating the TR-FRET signal from the donor (or acceptor) signal OR there is infinite positive co-operativity for binding of the nanobodies (both of them) to each receptor homo-dimer. There is no negative control, that demonstrates that when a donor and acceptor nanobody are bound to the outside of a cell at a receptor (or pair of cell surface proteins) that is known not to dimerize no TR-FRET (or at least substantially lower TR-FRET) signal is detected. In panel D, I agree that under saturating nanobody concentrations one would expect a hyperbolic TR-FRET signal from random collision, please provide data that demonstrates that in your system, with a non-dimerizing membrane protein you do indeed see this. This leaves the remainder of this figure uninterpretable. In the related figure S12, mGluR2 and a corresponding anti-mGluR2 nanobody are used to support TR-FRET specificity. This claimed specificity is not well supported by this data; 1. even though this is normalised data, the window for specific TR-FRET in this data set appears to be much, much smaller than in all other figures, 2. Where there is meant to be positive TR-FRET signal (2nd to last column C-E) that donor signal is higher (should go down) than in the control (last column), 3. Should be performed in saturation mode as in main figure 1. The data analysis in this panel is also inappropriate, it is clear that all data in C-E have been normalised to column 1 and this has been set to 100% for each independent experiment. There is therefore no variance in column 1, so this column cannot be used in an ANOVA.

Figure 2

Where is the quantification of data shown in panels A-C? Where are the controls (see above)?

Figure 3

Same comments as above: in the Raji, Jurkat and CCRF cell lines the TR-FRET signal should NOT track with total

fluorescence intensity, but rather should exhibit a sigmoidal curve.

Figure 4

Saturation data are required and again, donor intensity increases where TR-FRET is present and decreases where (theoretically) is absent (see above).

Figure S3

Please show data which represents the mean of separate biological experiments rather than technical replicates. Also please express the data as mean (biological replicates) with SD. Please state in the methods what model was used to fit the data shown in A and B.

Figure S4

This experimental set is predicated on the idea that the baseline BRET signal is a consequence of homo-oligomerization of either CXCR4 or ACKR3. Without an appropriate control of a receptor known not to oligomerize (or pair of plasma membrane proteins that do not interact) or a saturation BRET experiment (here the interpretation will be difficult) there is no justification for this assumption. The conclusion that CXCL12 induced increase in BRET is a consequence of dimerization is only one of several possible interpretations. For example, the increase in BRET could be a result of an increase in numbers of receptors at the cell surface or an increase in apparent receptor concentration due to some internalisation process – neither of which necessitate a change in oligomerization state. That the antagonist nanobody does not have this effect would be expected if these example alternative models were at play. I would suggest that either one of these models may be involved as the apparent potency for CXCL12 induced change in BRET is 4 orders of magnitude to the left of the apparent affinity OR that the reported CXCL12 effect is unspecific. In terms of interpretation of the lack of effect of either nanobody, all one can say is that an apparent change in oligomerisation state could not be detected, it is incorrect to say that they do not change the oligomerization state.

Figure S10/table S3

A. please show mean of independent experiments with SD. If ligand binding is a consequence of mass-action (which is an assumption of the model used to fit the data) then the error in the data should be log-normally distributed rather than normally distributed (as shown).

Throughout *, **, ***, **** are used to indicate different levels of significance. While this is still common practice this demonstrates a misunderstanding of statistical testing and should be discouraged.

Throughout SEM is used rather than SD, again this is common practice, however biological population variance is much more important than the precision with which a mean is known and I would strongly encourage the authors to use an estimate of variance.

Reviewer #3

(Remarks to the Author)

This manuscript describes the results obtained with a new method to evaluate GPCR oligomerization by applying nanobody labeling combined with TR-FRET to reveal the presence of endogenous CXCR4 oligomers both in artificial systems (in co-transfected cells and human cell lines with endogenous expression of CXCR4) and, more importantly, in native conditions (human peripheral blood mononuclear cells).

The authors first performed a series of additional control experiments to demonstrate that the previously characterized CXCR4-specific nanobody VUN400 and their developed ACKR3-specific nanobody VUN700 do not affect oligomerization and that their fluorescence labelling does not affect their binding affinities. The same as the previously reported antagonistic effect of VUN400, VUN700 antagonized the binding and signaling of the endogenous chemokine ligand CXCL12. Second, they performed carefully designed TR-FRET experiments with FRET donor and acceptor fluorophores attached to VUN400 and/or VUN700 in mammalian cells (CHO or HEK293T cells) transfected with CXCR4 and/or ACKR3, which supported the formation of homomers and heteromers and a preferential intracellular localization of ACKR3 oligomers compatible with constitutive internalization of the fluorescently labelled VUN700 bound to ACKR3. This was supported by showing a preferential plasma membrane localization of the VUN700 bound to a C-terminal-truncated ACKR3 devoid of internalization. Third, with the same method, they were able to detect endogenous CXCR4 homomers in several human leukemia/lymphoma-derived cell lines showing a significant expression of CXCR4, but not ACKR3. Finally, using the same method, they were able to detect CXCR4 homomers in native conditions, in human peripheral blood mononuclear cells.

The paper is well written, the experiments well designed and the results of importance to the field, providing a new method to demonstrate and study GPCR homomers and heteromers in native conditions. I do not have major or minor comments and recommend acceptance of the manuscript in its present form.

Version 1:

Reviewer comments:

Reviewer #1

(Remarks to the Author)

Thank you to the authors for addressing all my comments. I have a few minor additional comments relating to the resubmission.

Figure 1, 3 and Table 1 – I understand the author's explanation as to why referring to the 50% binding values as K_d is not correct. But using EC_{50} is also misleading. An EC_{50} is the half maximal effective concentration, which describes the concentration that produces a half-maximal functional response. It has nothing to do with binding. The authors should either use K_{app} (as they have mentioned) or something that indicates it is 50% bound receptors, e.g. $50\%B_{max}$.

Figure S13 – final paragraph, "very linearly" should be "vary linearly".

Line 246 – Figure S13A is referenced – I think this is the wrong figure being referred to.

Line 298 – "No measurable or a weak expression nor oligomers of ACKR3-WT" wording is unclear. Do the authors mean "No measurable or a weak expression of oligomers of ACKR3-WT"?

Reviewer #2

(Remarks to the Author)

I would like to thank the authors for their diligent responses to my comments. There are, however three issues that remain of concern to me. The first, I consider critical in terms of addressing the contention that dimers can be detected in vivo using the described reagents.

The second has been dealt with by the authors. On balance, their data supports their contention of being able to detect dimers in a recombinant setting. I would expect, however, that many GPCR experts will have reservations about the authors interpretation of this data. I suggest an alternative experiment, that would substantially strengthen the manuscript.

Lastly, I find the authors argument about statistical representation of the data puzzling and provide a more extensive exposition on why a different approach would be useful.

The first major additional experiment I requested was to demonstrate specificity of these nanobodies against other receptors. Minimally this needs to include cross reactivity against related chemokine receptors including the CXCR family (1,2,3,5, and 6) and ACKR family (1,2, and 4). Without this, there is no legitimate claim on being able to detect dimers of the target receptors in vivo. Please perform appropriate specificity tests. The whole premise of the manuscript hangs on the specificity of these reagents in a native system.

I requested that a control be included that demonstrates no TR-FRET between non-dimerizing cell-surface proteins, I apologise for not noticing the data presented in S14 (formerly S12). Can I suggest that this data be moved to a main figure as this is an important control? Can the authors please comment on what they believe the proximity is likely to be between DN1-d2 and VUN400Tb, in the case that mGluR2 (already a dimer) and CXCR4 should they be in a complex (tetrameric)? Based on the published structure of mGluR5 and the binding site of DN1-d2, there would be at least another 12 nm (possible much more) added to distance between donor and acceptor. Is the proximity here close enough, that any signal from random collision would still be evident? I am concerned about interpretation of saturation TR-FRET using either a soluble (bulk solution) or lipophilic (membrane preferring) fluorescein to support the notion that random collision will produce a hyperbolic curve. This is indeed what they observe, but may be open to interpretation. They compare the hyperbolic curves (Fig S13) to linear plots (Fig 1D) assuming that the same ranges of concentration can be tested when transfecting a receptor and having this express at the cell-surface and adding a soluble fluorescent dye. If one examines fig S13, then it appears that if the first ~ 6 points were replotted without the highest concentrations this would be a good linear fit (as seen in Fig 1D for the receptor). One solution to the above difficulties in terms of interpretation would be to mutate CXCR4 to prevent dimerization and use this as a negative control. A dimeric structure for CXCR4 is published (PDB: 3OEO, doi.org/10.1126/science.1194396) from which it should be possible to design mutations that would preserve function/expression whilst blocking dimerization.

Comments on statistics:

There still appear to be ANOVAs that are performed where one group has been normalised to 100%, thus removing the variance of this group. This is not appropriate.

Once again, I acknowledge the sloppy practice of other researchers in reporting SEM rather than SD and the use of different numbers of * to indicate different P values. With reference to SEM, I can only assume that when the authors walk in to their doctor and have their blood pressure taken, they must immediately call the ambulance when they find their diastolic is outside the population SEM..... for any well-trained biologist, an estimate of the population variance (i.e. SD) of any biological parameter is much more important than any measure of how accurately one has estimated the population mean (S.E.M.) - see references below. With respect to using different numbers of * to indicate different levels of statistical significance; this appears to indicate two things, 1, that the level of significance deemed to be appropriate has been decided post-hoc (a type of P hacking), and 2, that the authors believe that a lower P-value is related to a change in biological significance (untrue). I would suggest the following articles:

Michel, M. C., Murphy, T. J., & Motulsky, H. J. (2020). New Author Guidelines for Displaying Data and Reporting Data

Analysis and Statistical Methods in Experimental Biology. *Molecular pharmacology*, 97(1), 49–60.

<https://doi.org/10.1124/mol.119.118927>

Halsey, L. G., Curran-Everett, D., Vowler, S. L., & Drummond, G. B. (2015). The fickle P value generates irreproducible results. *Nature methods*, 12(3), 179–185. <https://doi.org/10.1038/nmeth.3288>

Ronald L. Wasserstein & Nicole A. Lazar (2016) The ASA's Statement on p-Values: Context, Process, and Purpose, *The American Statistician*, 70:2, 129-133, DOI:10.1080/00031305.2016.1154108

Curran-Everett, D., & Benos, D. J. (2004). Guidelines for reporting statistics in journals published by the American Physiological Society. *American journal of physiology. Regulatory, integrative and comparative physiology*, 287(2), R247–R249. <https://doi.org/10.1152/ajpregu.00346.2004>

Curran-Everett, D., & Benos, D. J. (2007). Guidelines for reporting statistics in journals published by the American Physiological Society: the sequel. *Advances in physiology education*, 31(4), 295–298.

<https://doi.org/10.1152/advan.00022.2007>

Version 3:

Reviewer comments:

Reviewer #2

(Remarks to the Author)

I am pleased to see the additional specificity data provided in S11 panel C. This addresses the most critical issue regarding the ability of the results from transfected cell studies to be translated to endogenous settings with some degree of confidence.

I accept the difficulty with performing additional requested experiments. Having the most important specificity issue addressed is fine.

The authors appear to have changed the error bars in figure S11, S13 and S14 to SD. In their response they state that "All results are now presented with standard deviation rather than SEM" - this does not appear to be the case and all the figure legends (apart from those noted above) still state SEM. I don't know if this was perhaps a problem with not uploading the corrected figures and legends?

Statistics remain invalid for S14 due to normalization, non-parametric test such as Kurskal Wallis would address variance issue in control but would not fix the experimental design. Suggest doing statistics on non-normalised (raw data). Elsewhere statistics sill have issues.

Rebuttal letter

Reviewer #1 (Remarks to the Author):

This a novel study describing the use of nanobodies to investigate GPCR oligomers. Overall, the study is very interesting, and makes a great contribution to the field. The paper itself requires a little more work till it is publication ready.

We thank reviewer 1 for his/her constructive and very meticulous review. All the suggestions have been followed.

There is a Table S1 and a Table S2 but no Table S3?

The reviewer is right, the text has been modified and Table S3 has been renamed Table S2.

The authors should conduct some experiments to confirm the selectivity of the nanobodies. Does VUN-400 bind to ACKR3 and does VUN-700 bind to CXCR4?

Reviewer 1 is right and we performed saturation experiments with VUN400-d2 and VUN700-d2 on both receptors HALO-CXCR4 and SNAP-ACKR3 labelled with Lumi4-Tb. The results are presented in Figure S11 (see below) clearly indicated that VUN400-d2 but not VUN700-d2 can bind to HALO-CXCR4. By contrast VUN700-d2 but not VUN400-d2 can bind to SNAP-ACKR3, proving the selectivity of both VUN400 and VUN700 for their cognate receptor.

Can the authors please explain how three unlabelled ligands (unlabelled VUN700, CXCL12, and VUF15485 – from line 629 and in Fig S11) have been used to generate the specific binding curve shown in Fig 1B? Usually only a one unlabelled ligand would be used to generate specific binding, so this doesn't make sense.

We modified the legend of Figure 1 and Figure S12 (Figure S11 in the previous version) to make it clearer. Specific binding signals were obtained by subtracting non-specific binding signals obtained in the presence of an excess of unlabeled competitor from the total binding. For ACKR3, we used a mixture of three unlabeled competitors: VUN700, CXCL12 and VUF15485.

Fig 1 legend – some of the wording for B is in A, i.e. talking about ACKR3 in A, but that should be in B.

Table 1 does not appear to have any EC50 values, only Kd values, even though Line 155 and the table legend mentions EC50s. I think the authors mean to say Kd Values?

Reviewer 1 is correct and we modified consequently the legend. ACKR3 has been removed from legend A) and added to legend B).

Regarding Table 1, Reviewer is right: because the saturation curves in Figure 1 have been obtained with a mix of nanobodies labeled either with Lumi4-Tb or d2 at equimolar concentrations, the EC50 does not correspond to the Kd but to an apparent Kd ($K_{d_{app}}$). Therefore, to avoid any confusion, we replaced “Kd” with “EC50” to in Table 1.

Line 162 – “The maximum corresponding” would be clearer as “The maximum signal corresponding”

The sentence has been modified according to Reviewer 1 suggestion.

Line 172 – “Because random collisions occurrence varies as a power two of the receptor density, plotting TR-FRET as a function of receptor density would have led to a parabolic curve⁴²” I think this needs more clarification, I can’t see in the referenced article where it says a linear curve indicates specific interactions while parabolic curve indicates random collisions.

Thanks to Reviewer 1 to raise this point.

Indeed, we observed a linear variation of TR-FRET as a function of receptor density. This is not compatible with random collisions of receptors diffusing in the membrane since random collisions leads to parabolic TR-FRET curve when partner density is increased. To illustrate our point, we performed additional TR-FRET experiments with a “pure” collision model. We expressed HALO-CXCR4 receptor in CHO-K1 cells, labelled them with Lumi4-Tb and incubated cells in the presence of increasing concentration of fluorescein. TR-FRET was plotted as a function of fluorescein concentration and it followed a parabolic curve. This is illustrated in Figures S13 (see Figure S13 bellow).

Moreover we changed the reference since information were in the supplementary information and are not so clear. We referred in the present version to Maurel et al, Nature Method 2008.

We modified the text in the manuscript as followed:

“The TR-FRET as a function of the receptor density varied linearly (Figure 1D, R^2 values >0.8) by contrast to TR-FRET resulting from random collision (see Figure S13). This indicates that TR-FRET efficiency within CXCR4 receptors is constant, not dependent on receptor density and therefore did not result from random collision of receptors diffusing in the membrane⁴²”

Figure S12 – this is a very confusing figure. The figure legend needs to be reworded substantially. Specifically, it would be better to move this description of the graphs to the beginning of the

legend:

“Transfected CHO cells similarly expressing HALO-CXCR4 and SNAP-mGLUR2 receptors (B) are incubated in the presence of a mix of VUN400-Tb, VUN400-d2 and DN1-d2 nanobodies⁴, the latter targeting mGluR2 receptors (A). Donor (C), acceptor (D) and TR-FRET (E) luminescence signals are recorded. Despite significant labeling with VUN400-Tb (C) and/or DN1-d2 (D), no significant specific TR-FRET signal is observable between the nanobodies (E). By contrast, while the labeling with VUN400-d2 is much less intense than that observed with DN1-d2 (D), specific TR-FRET is observed between VUN400-Tb and VUN400-d2 (E). These data strongly suggest that TR-FRET signal is due to the existence of specific interactions between CXCR4 receptors and not to random collisions between receptors.”

By having it at the beginning of the figure legend, the reader understands immediately what the figure is about. Currently, less important details about the figure come first, which makes the legend confusing to read. Also, I assume that in panel A, the blue receptors are CXCR4 and the green ones are mGluR2, but again, this isn't specified in the legend, which it should be. Panel A also needs more explanation.

Many thanks for the constructive suggestion. We modified the legend accordingly. We also specified the color code for panel A. The figure has been renumbered to S14.

Figure 2D – the graph in this panel needs more explanation. How are the specific binding signal and the total binding signal determined in this graph? It is not stated in the figure legend.

The reviewer is correct and we apologize for the lack of sufficient explanation. Since the panel D was not so informative, we chose to replace it with plots showing the quantification the different signals. Luminescence of donor (blue line), fluorescence of acceptor (red line) and TR-FRET signal (magenta line) along the green lines on the images. In panel C, the quantification clearly shows that in a mixture of cells expressing either HALO-CXCR4 or SNAP-ACKR3-ΔCterm, the TR-FRET signal (magenta) is not significant. The legend of the figure 2 has been modified accordingly.

Figure 3 – I think it would be good to compare the overall FRET values in these graphs. I assume C and D would have lower FRET values than E, F, G, but because all the data is normalised, that comparison can't be made. Also, as in Fig 2, how are the specific binding signal and the total binding signal determined in this graph? It is not stated in the figure legend. Also, Fig 3 D is only n=2. Presumably this should be n=3.

Reviewer 1 is right, FRET intensities in the experiments shown in panels C and D were indeed lower than in the other panels. The reason for this is shown in panel A in which the CXCR4 density in the different cell lines has been measured by flow cytometry. Because we want to emphasize the shape of the curves, we chose to present the donor and acceptor fluorescence and TR-FRET signals as a percentage of the maximum reached at 30 nM to pool the different experiments. The TR-FRET signals in panels C and D, corresponding to THP1 and HEK293 cell lines, respectively, are not significantly higher than background. The TR-FRET intensities for panels E, F and G are in the same range.

We modified the legend to specify how we got the non-specific signal (in the presence of an excess of IT1t (10 μM): “The specific binding signal was calculated by subtracting the non-specific binding signal determined in the presence of an excess of IT1t (10 μM), from the total binding signal.”

Review: specific comments

Line 47 – “Nevertheless, oligomerization of GPCRs was mostly investigated” should be “Nevertheless, oligomerization of GPCRs has mostly been investigated”.

The sentence has been modified as suggested.

Line 50 – “previously referred to as CXCR7” - I would put this phrase in brackets rather than commas, as it currently reads a bit ambiguously whether “previously referred to as CXCR7” refers to just ACKR3 or both CXCR4 and ACKR3.

We agree with this suggestion. The brackets have been added and commas suppressed. **Line 134 – “resulted” should be “resultant”.**

The sentence has been modified.

Figure S12B – one of the error bars appears to be white so is not visible. It should be changed to black like all the other error bars.

Figure S14 (S12 in the previous version) has been modified as suggested. The error bars has been changed to black.

Line 248 – “CXCR4 oligomers in cancer cell line” should be “CXCR4 oligomers in cancer cell lines”
“Cancer cell line” has been changed to “cancer cell lines”.

Line 250 – “and especially” – “specifically” would be more appropriate wording here.

The sentence has been modified.

Line 277 – “directed against” should be “directed them against”

he sentence has been modified.

Line 297 – “vary” should be “variation”

Indeed, “variation” has been substituted to “vary”.

Supp Methods:

Second paragraph, second line – “CHO-K1 cells. Cells were” should be “CHO-K1 cells were”.

It has been corrected.

Fourth paragraph, third line – “method 7 Briefly” should be “method 7. Briefly”.

The correction has been made.

Fifth paragraph, third line – “refer to Tables2” should be “refer to Table S1”.

The correction has been made.

Reviewer #2 (Remarks to the Author):

This manuscript describes the development of CXCR4 and ACKR3 directed nanobodies, their labelling and assessment of the ability of these nanobodies to detect receptor oligomerization, with the intent to be able to detect this in natively expressing systems. This is a very good idea because, as the authors point out, dimerization of GPCRs remains controversial and there are no reliable data (that I'm aware of) to support dimerization in native settings. Unfortunately, this is let down by the poor design of experiments, lack of controls, inappropriate analysis and over-interpretation. As a consequence, there are no conclusions that are drawn that can be adequately supported by the presented data set.

The specificity of the nanobodies used is only validated in CHO cells through competition using CXCL12. These nanobodies may still recognise other receptors/proteins (in particular other chemokine receptors) that are not expressed on these cells. The contention of the manuscript is that the nanobodies will be able to be used to detect oligomerization in native tissues. Minimally this needs to be supported by data that demonstrates these are not cross reactive with closely related chemokine receptors. Ideally, binding to PBMC where their relevant targets have been knocked out would make sense.

Reviewer 2 is correct and this was also requested by reviewer 1. We evaluated included an extra experiment showing the selectivity of the nanobodies VUN400 and VUN700. Since both CXCR4 and ACKR3 receptors were able to bind CXCL12, we performed saturation experiments with VUN400-d2 and VUN700-d2 on both HALO-CXCR4 and SNAP-ACKR3 receptors. The results are presented in Supplementary Figure S11 (see below). It clearly indicates that VUN400-d2 and VUN700-d2 can bind only on HALO-CXCR4 and SNAP-ACKR3, respectively. No cross-reaction has been observed proving the selectivity of both nanobodies.

Figure 1/table 1.

The apparent affinity of all antibodies in this data set are almost a full log lower than in the taglite assay – this needs to be addressed.

First, we performed additional experiments to carefully assess the K_d of VUN400-d2 and VUN700-d2. We found that these are slightly greater than the ones indicated in the publication. We indicated the

new values in Table 1 and indicated in the legend. Differences in binding affinities of the nanobodies used in this study and previously published variants thereof are likely caused by the different tag used in this study. Although of less impact than N-terminal modifications, adjustment of the C-terminal tag can slightly affect the affinity of nanobodies.

Second, the setup of the binding experiments corresponding to the values reported in Table 1 did not correspond to “classic” saturation or competitive experiments. In the binding experiment, both nanobodies, VUN400 and VUN700 were added at the same concentration. The legend in the previous version (Kd) was therefore not adapted and we changed it for p(EC50).

In the data fit, the data are expressed as %max, yet the max will clearly be over 100%, which makes no sense.

The legend of the Y axis was indeed ambiguous. We therefore changed it for “intensity (% of the intensity at 30 nM)”, 30 nM corresponding to highest concentration of the nanobodies we used. Of note, we did not draw any conclusion on the intensity but only on the EC50 of the curve, which is independent of the normalization. Therefore, the conclusions did not change.

In the table affinities are reported with linear error, data is reported as being fitted using a 1-site binding model in PRISM, this model assumes mass-action and therefore error should be log normally distributed. The data does not make any sense, whatsoever.

Reviewer 2 is right. We changed the values in the different table, which are now indicated as p(EC50) or pKd, errors were calculated from the log values.

The TR-FRET signal tracks perfectly with fluorescence intensity from the individual antibodies. At concentrations below full occupancy, and in particular at concentrations below 50% occupancy, one would expect a mixture of singly occupied dimers and doubly occupied dimers. This will result in a sigmoidal TR-FRET signal. The only ways I can envision getting the result that is reported is either the filters used are not actually separating the TR-FRET signal from the donor (or acceptor) signal OR there is infinite positive co-operativity for binding of the nanobodies (both of them) to each receptor homo-dimer.

Thanks to the Reviewer 2 for his comment. In the former version, we wrote that EC50 values of the different curves were in the same range without discussing into details. For CXCR4, the curves are superimposed while, for ACKR3, a right shift is observed for the TR-FRET curve compared to the other two.

Regarding the separation of the donor/acceptor and TR-FRET signals, Lumi4-Tb indeed exhibits a small luminescence at 665 nm, the wavelength at which TR-FRET is measured. However, this signal is negligible at low concentration. Moreover, because of an excitation at 337 nm and the introduction of time delay (50 μ s), emission of d2 resulting from direct excitation of d2 (acceptor) is also negligible.

How to explain our results? We used a simplified theoretical model derived from the one we previously established (Durroux et al, TiPS 2005) to model a binding curve when considering a monomer/dimer equilibrium. To simplify the model, we applied three assumptions: i) nanobody labeling does not significantly impact nanobody affinities for the receptor. Therefore, the nanobodies either with Lumi4-Tb or d2 have the same affinity; ii) we considered that the affinity of a nanobody on a monomeric receptor is equal to the one of the first nanobody on a dimeric form ($\alpha=1$ in the

table below); iii) we considered that both labelled nanobodies were added at the same concentration, as we did in the experimental conditions corresponding to Figure 1.

The results presented below correspond to four different hypothetical situations:

- **A:** two nanobodies can bind to a dimer with the same affinity (no positive or negative cooperative binding) (see table A below: $\alpha=\beta=\gamma=1$). We fixed the affinity to 5 nM (approximately the affinity we determined for the nanobodies with the Taglite assay). Moreover, the constant of equilibrium between monomer and dimer (K_{dim}) was set to 0.1 nM. At this value, the proportion of monomer in the absence of nanobodies is about 98% of the total receptor.
- **B:** there is a positive cooperative binding of nanobodies on a dimer. We considered a cooperative factor of 0.1. It means that the affinity of a second nanobody is 10 times better when a first nanobody is already bound to the receptor (see Table A: $\alpha=1$; $\beta=\gamma=0.1$). As in condition A, K_{dim} was set at 0.1 nM. Of note, positive cooperativity in the binding has been reported for antagonists on different receptors (for example see Matera et al., 1985)
- **C:** positive cooperativity for the binding of a second nanobody when a first nanobody is already bound and we set K_{dim} at 0.01 nM, meaning that in the absence of nanobodies, the proportion of monomer is about 85% of the total receptor.
- **D:** positive cooperativity for the binding of a second nanobody when a first nanobody is already bound and we set K_{dim} at 0.001 nM, meaning that in the absence of nanobodies, the proportion of monomer is about 50% of the total receptor.

All parameter values used for the models are indicated in the table below.

	A	B	C	D
Rt (M)	1 E-12	1 E-12	1 E-12	1 E-12
K_{dim} (M)	1E-10	1E-10	1E-11	1 E-12
Kd (M)	5 E-9	5 E-9	5 E-9	5 E-9
α	1	1	1	1
β	1	0.1	0.1	0.1
γ	1	0.1	0.1	0.1

With $K_{dim} = [R]^2/[RR]$

$Kd = [N1][R]/[RR] = [N2][R]/[RR]$

$\alpha Kd = [N1][RR]/ [N1RR] = [N2][RR]/ [N2RR]$

$\beta Kd = [N1][N1RR]/ [N1RRN1] = [N1][RRN1]/ [N1RRN1]$
 $= [N2][N2RR]/ [N2RRN2] = [N2][RRN2]/ [N2RRN2]$

$\gamma Kd = [N2][N1RR]/ [N1RRN2] = [N2][RRN1]/ [N2RRN1]$

Rt : total receptor (1e-12 M); N1: Nanobody 1; N2: Nanobody 2; K_{dim} : dissociation constant of dimer, α , β and γ correspond to the cooperative coefficients for the binding of nanobodies on the different dimer/ nanobody complexes.

The different binding curves are illustrated in the figure below. Insets correspond to a zoom on the feet of the curves.

Curves were fitted when considering one-site saturation equation in Prism although we are aware that it does not exactly correspond to the experimental conditions used for Figure 1 of the manuscript. EC50 values were reported in the table below.

	EC50		
	Tb	D2 fluorescence	TR-FRET
A	2.50	2.50	7.30
B	2.36	2.36	5.84
C	1.83	1.83	3.26
D	1.25	1.25	1.88

In condition A, we observed a right shift of the TR-FRET curve; the EC50 for TR-FRET signal is about 3 times higher than for those of Tb and d2 fluorescence signals. The differences between the EC50s are decreased when considering a positive cooperative binding on dimeric receptor (condition B) et even more when considering the presence of a significant percentage of dimer in the absence of nanobodies (Conditions C and D).

Moreover, all the insets clearly show sigmoid curves but sigmoidity of the curves can be observed only on a narrow range of nanobody concentration.

Did our results for CXCR4 make sense with these models?

Indeed, for CXCR4, we observed superimposed curves. This could suggest the presence of a substantial % of dimeric receptors in the experimental conditions used to express CXCR4 receptors (high expression of HALO-CXCR4 in CHO or HEK 293 cells). This could be in accordance with the results reported by M. Lohse research group which has reported that when CXCR4 were expressed at high concentration, CXCR4 homodimers became prevalent (Isbilir et al, PNAS 2023).

Regarding the sigmoidity of the curves, because it can only be observed on a narrow range of nanobody concentrations, it is almost impossible to observe it with the Tag lite binding assay.

Therefore, we modified the text in the new version as followed:

“It is noteworthy that a rightward shift and sigmoidal curves would have been expected for the TR-FRET signal, but such a pattern may be difficult to discern if a large proportion of the receptors are dimeric, as has been reported for CXCR4 when expressed at high density.²¹ (see supplementary Information – Binding curve modelling section).”

The binding curve modeling and the discussion have been added in the Supplementary Information.

There is no negative control, that demonstrates that when a donor and acceptor nanobody are bound to the outside of a cell at a receptor (or pair of cell surface proteins) that is known not to dimerize no TR-FRET (or at least substantially lower TR-FRET) signal is detected. In panel D, I agree that under saturating nanobody concentrations one would expect a hyperbolic TR-FRET signal from random collision, please provide data that demonstrates that in your system, with a non-dimerizing membrane protein you do indeed see this. This leaves the remainder of this figure uninterpretable. In the related figure S12, mGluR2 and a corresponding anti-mGluR2 nanobody are used to support TR-FRET specificity. This claimed specificity is not well supported by this data; 1. even though this is normalised data, the window for specific TR-FRET in this data set appears to be much, much smaller than in all other figures, 2. Where there is meant to be positive TR-FRET signal (2nd to last column C-E) that donor signal is higher (should go down) than in the control (last column), 3. Should be performed in saturation mode as in main figure 1. The data analysis in this panel is also inappropriate, it is clear that all data in C-E have been normalised to column 1 and this has been set to 100% for each independent experiment. There is therefore no variance in column 1, so this column cannot be used in an ANOVA.

First, Regarding the TR-FRET signal variation as a function of partner density, we performed additional experiments. To be sure to consider a pure “random collision” molecular model and to keep the expression of CXCR4 constant, we have transfected CHO-K1 cell with HALO-CXCR4. After labelling the cells with Lumi4-Tb, we incubated cells in the presence of increasing concentrations of fluorescein. The variation of the TR-FRET signal was plotted as a function of fluorescein concentration. Results are now illustrated in Figure S13 (see below). The text of the manuscript have been modified as followed :

“The TR-FRET as a function of the receptor density varied linearly (Figure 1D, R^2 values >0.8) by contrast to TR-FRET resulting from random collision (see Figure S13). This indicates that TR-FRET efficiency within CXCR4 receptors is constant, not dependent on receptor density and therefore did not result from random collision of receptors diffusing in the membrane⁴²”

Figure S13: Variation of TR-FRET signal resulting from random collisions between labelled partner follows a parabolic curve. Two sets of experiments were performed to study the variation of TR-FRET signal resulting from a model of pure random collisions between two partners. In a first set of experiments, transfected CHO-K1 cells expressing HALO-CXCR4 were labelled with Lumi4-Tb and incubated in the presence of increasing concentration of fluorescein (green circle) in the extracellular medium (A). In a second set of experiments, transfected CHO-K1 cells expressing HALO-CXCR4 were labelled with Lumi4-Tb and incubated in the presence of increasing concentration of MemBright 488 (green circle), a fluorescent dye which is incorporated into the membrane (B). (concentration of 1 corresponds to the recommended concentration by supplier) TR-FRET signal (520 nm) was plotted as a function of dye concentration. Data were expressed as mean + sem were obtained from three independent experiments performed in quadruplicate.

Both plots illustrated that TR-FRET signal does not vary very linearly in function of the concentration of the partners and the TR-FRET efficiency is dependent on the concentrations of the partner.

Moreover, Reviewer 2 mentioned that we did not provide negative control with receptors which are not supposed to dimerize. Although the control is not illustrated in the main text, it is in the supplementary information (Fig. S12, now S14) as mentioned by Reviewer 2. Reviewer 2 made three comments:

-1- even though this is normalised data, the window for specific TR-FRET in this data set appears to be much, much smaller than in all other figures

Although it is not clear what Reviewer 2 means with “much, much smaller”, it is indeed smaller than in the other figures. The receptor density on the cell surface can be 7- to 10-fold higher when CXCR4 is expressed alone. But this density decreases when CXCR4 is co-expressed with another receptor. In the present study, the expression levels of CXCR4 and mGluR2 when co-expressed, are the same range as of CXCR4 and ACKR3 when co-expressed. Moreover, we have chosen conditions in which both receptors are expressed at approximately the same cell surface density. Detecting an absence

of significant TR-FRET signals using an unbalanced receptor expression instead, would have not been conclusive. Finally, despite the lower receptor density of CXCR4, the TR-FRET signal corresponding to CXCR4 labelling is still significant. This is not the case for the negative control with CXCR4 and mGluR2.

-2- Where there is meant to be positive TR-FRET signal (2nd to last column C-E) that donor signal is higher (should go down) than in the control (last column),

We understand the comment of Reviewer 2 and this would indeed be the case if the labelling of CXCR4 was constant between control and positive TR-FRET conditions. The control condition was in the presence of a large excess of IT1t, a competitor of VUN400-Tb for the binding to CXCR4. Therefore, in these conditions, the binding of VUN400-Tb is reduced, and, as expected, this also results in lower donor signals (**after the washing steps performed before luminescence measurements**, as indicated in Material and method section). The results would have been different in the absence of washing.

-3- . Should be performed in saturation mode as in main figure 1. The data analysis in this panel is also inappropriate, it is clear that all data in C-E have been normalised to column 1 and this has been set to 100% for each independent experiment. There is therefore no variance in column 1, so this column cannot be used in an ANOVA.

Referee 2 is right and the figure has been modified.

Figure 2

Where is the quantification of data shown in panels A-C? Where are the controls (see above)?

Quantification of the different signal along an axis passing through the cells (green line) have now been added to the right of each panel. Blue, red and magenta lines correspond to donor, acceptor and TR-FRET signal, respectively.

Figure 3

Same comments as above: in the Raji, Jurkat and CCRF cell lines the TR-FRET signal should NOT track with total fluorescence intensity, but rather should exhibit a sigmoidal curve.

See comment above and Supplementary Information (Binding curve modeling section).

Figure 4

Saturation data are required and again, donor intensity increases where TR-FRET is present and decreases where (theoretically) is absent (see above).

We agree, saturation experiments would have been better. But unfortunately, this is simply not achievable on PBMC considering the number of cells needed for one experiment. Considering the TR-FRET signal, because IT1t prevents the binding of the nanobodies, weaker signals of the donor were obtained (see above).

Figure S3

Please show data which represents the mean of separate biological experiments rather than technical replicates. Also please express the data as mean (biological replicates) with SD. Please state in the methods what model was used to fit the data shown in A and B.

Reviewer 2 is right and we apologize for the mistake. Results corresponded to three independent experiments, each performed in triplicate and therefore we modified the legend.

We also indicated in the legend the name of the subroutine of Prism used to fit the data of panel A and B

Figure S4

This experimental set is predicated on the idea that the baseline BRET signal is a consequence of homo-oligomerization of either CXCR4 or ACKR3. Without an appropriate control of a receptor known not to oligomerize (or pair of plasma membrane proteins that do not interact) or a saturation BRET experiment (here the interpretation will be difficult) there is no justification for this assumption. The conclusion that CXCL12 induced increase in BRET is a consequence of dimerization is only one of several possible interpretations. For example, the increase in BRET could be a result of an increase in numbers of receptors at the cell surface or an increase in apparent receptor concentration due to some internalisation process – neither of which necessitate a change in oligomerization state. That the antagonist nanobody does not have this effect would be expected if these example alternative models were at play. I would suggest that either one of these models may be involved as the apparent potency for CXCL12 induced change in BRET is 4 orders of magnitude to the left of the apparent affinity OR that the reported CXCL12 effect is unspecific. In terms of interpretation of the lack of effect of either nanobody, all one can say is that an apparent change in oligomerisation state could not be detected, it is incorrect to say that they do not change the oligomerization state.

Different studies have reported CXCR4 and ACKR3 oligomerization using resonance energy transfer strategies and in particular BRET strategies (Babcock et al, JBC 2003, Pecherancier et al JBC 2005, Armando et al, Faseb J., 2014). Does the FRET signal only dependent on the receptor oligomerization? Most probably not and non-specific BRET signal could be responsible for part of the signal. This non-specific signal can be due to stochastic collisions of receptors diffusing into the membrane and it could decrease or increase when receptor density is affected during receptor targeting of receptor internalization.

In the experiment we have reported, we did not draw any conclusion on the receptor oligomerization and Reviewer 2 is right, different cellular processes can be responsible of the variation of the BRET signal. We simply showed that BRET signal can be modulated by CXCL12. By contrast, VUN400 did not impact BRET signal, suggesting that oligomerization is not affected. If variations of the BRET signal would have been modified, we would not be able to conclude to the impact of ligands on receptor oligomerization but the absence of BRET signal modification is a strong argument to conclude to the absence of effect of nanobodies on CXCR4 oligomerization.

Figure S10/table S3

A. please show mean of independent experiments with SD. If ligand binding is a consequence of mass-action (which is an assumption of the model used to fit the data) then the error in the data should be log-normally distributed rather than normally distributed (as shown).

Table S3, (named now Table S2) has been modified.

Throughout *, **, ***, * are used to indicate different levels of significance. While this is still common practice this demonstrates a misunderstanding of statistical testing and should be discouraged.***

Throughout SEM is used rather than SD, again this is common practice, however biological population variance is much more important than the precision with which a mean is known and I would strongly encourage the authors to use an estimate of variance.

*******, ********, *********, ********** are frequently used throughout scientific literature and in particular in Communication Biology. We do not see the problem when the significance of the symbols is indicated in the legend. Table values are now indicated with SD.

Reviewer #3 (Remarks to the Author):

This manuscript describes the results obtained with a new method to evaluate GPCR oligomerization by applying nanobody labeling combined with TR-FRET to reveal the presence of endogenous CXCR4 oligomers both in artificial systems (in co-transfected cells and human cell lines with endogenous expression of CXCR4) and, more importantly, in native conditions (human peripheral blood mononuclear cells).

The authors first performed a series of additional control experiments to demonstrate that the previously characterized CXCR4-specific nanobody VUN400 and their developed ACKR3-specific nanobody VUN700 do not affect oligomerization and that their fluorescence labelling does not affect their binding affinities. The same as the previously reported antagonistic effect of VUN400, VUN700 antagonized the binding and signaling of the endogenous chemokine ligand CXCL12. Second, they performed carefully designed TR-FRET experiments with FRET donor and acceptor fluorophores attached to VUN400 and/or VUN700 in mammalian cells (CHO or HEK293T cells) transfected with CXCR4 and/or ACKR3, which supported the formation of homomers and heteromers and a preferential intracellular localization of ACKR3 oligomers compatible with constitutive internalization of the fluorescently labelled VUN700 bound to ACKR3. This was supported by showing a preferential plasma membrane localization of the VUN700 bound to a C-terminal-truncated ACKR3 devoid of internalization. Third, with the same method, they were able to detect endogenous CXCR4 homomers in several human leukemia/lymphoma-derived cell lines showing a significant expression of CXCR4, but not ACKR3. Finally, using the same method, they were able to detect CXCR4 homomers in native conditions, in human peripheral blood mononuclear cells.

The paper is well written, the experiments well designed and the results of importance to the field, providing a new method to demonstrate and study GPCR homomers and heteromers in native conditions. I do not have major or minor comments and recommend acceptance of the manuscript in its present form.

Many thanks to Reviewer3 for his/her very positive comments.

Rebuttal letter

Answers to the Reviewers

Reviewer #1 (Remarks to the Author):

Thank you to the authors for addressing all my comments. I have a few minor additional comments relating to the resubmission.

Figure 1, 3 and Table 1 – I understand the author’s explanation as to why referring to the 50% binding values as K_d is not correct. But using EC_{50} is also misleading. An EC_{50} is the half maximal effective concentration, which describes the concentration that produces a half-maximal functional response. It has nothing to do with binding. The authors should either use K_{app} (as they have mentioned) or something that indicates it is 50% bound receptors, e.g. $50\%B_{max}$.

Good to hear that reviewer #1 appreciated us addressing all comments. We understand the reviewers comment on the use of the term EC_{50} in this case. We would suggest to use $TR-FRET_{50}$ which seems to be more appropriate.

Figure S13 – final paragraph, “very linearly” should be “vary linearly”.

We apologize for the spelling mistake, the suggested correction has been made.

Line 246 – Figure S13A is referenced – I think this is the wrong figure being referred to. Reviewer 1 is correct, we made the correction and referred now to Figure S15A

Line 298 – “No measurable or a weak expression nor oligomers of ACKR3-WT” wording is unclear. Do the authors mean “No measurable or a weak expression of oligomers of ACKR3-WT”?

The correction has been made

Reviewer #2 (Remarks to the Author):

I would like to thank the authors for their diligent responses to my comments. There are, however three issues that remain of concern to me. The first, I consider critical in terms of addressing the contention that dimers can be detected in vivo using the described reagents.

The second has been dealt with by the authors. On balance, their data supports their contention of being able to detect dimers in a recombinant setting. I would expect, however, that many GPCR experts will have reservations about the authors interpretation of this data. I suggest an alternative experiment, that would substantially strengthen the manuscript.

Lastly, I find the authors argument about statistical representation of the data puzzling and provide a more extensive exposition on why a different approach would be useful.

The first major additional experiment I requested was to demonstrate specificity of these nanobodies against other receptors. Minimally this needs to include cross reactivity against related chemokine receptors including the CXCR family (1,2,3,5, and 6) and ACKR family (1,2, and 4). Without this, there is no legitimate claim on being able to detect dimers of the target receptors in vivo. Please perform appropriate specificity tests. The whole premise of the manuscript hangs on the specificity of these reagents in a native system.

We thank reviewer #2 for noticing our efforts to respond to the first set of comments on our manuscript.

This point of the selectivity was rightly raised in the previous review by both Reviewer #1 and #2. In the previous version of the manuscript, we provided experiments which proved the selectivity of VUN400 and VUN700 on CXCR4 and ACKR3, respectively. Experiments were performed on these two receptors since they exhibit overlapping pharmacology. None of the receptors mentioned by Reviewer #2 (*CXCR family (1,2,3,5, and 6) and ACKR family (1,2, and 4)*) has been shown not to bind any ligand that binds to CXCR4 and ACKR3, (cf. IUPHAR website (<https://www.guidetopharmacology.org/GRAC/FamilyDisplayForward?familyId=14>)). Furthermore, ACKR2 and ACKR4 binds CCL-type chemokines and not CXCL chemokines. Moreover, Reviewer #2 wrote that “Without this, there is no legitimate claim on being able to detect dimers of the target receptors *in vivo*”. Therefore, we focused on CXCR4, the only receptor for which we concluded that oligomers exist in native tissues (*in vivo*). No conclusions could be drawn for ACKR3, as the signals were too weak.

However, as do see the additive value of some specificity testing, we now performed binding experiments with VUN400-d2 on SNAP-CXCR1/2/3/5/6 and SNAP/ACKR1 receptors. HALO-CXCR4 was used as positive control. In addition, an excess of unlabeled VUN400-VHH was added to determine the non-specific binding of VUN400-d2. Three independent experiments were performed in triplicate and results are presented in supplementary information file (Figure S11, panel C). The results clearly indicate the absence of cross reactivity of VUN400 with other receptors.

I requested that a control be included that demonstrates no TR-FRET between non-dimerizing cell-surface proteins, I apologise for not noticing the data presented in S14 (formerly S12). Can I suggest that this data be moved to a main figure as this is an important control? Can the authors please comment on what they believe the proximity is likely to be between DNI-d2 and VUN400Tb, in the case that mGluR2 (already a dimer) and CXCR4 should they be in a complex (tetrameric)? Based on the published structure of mGluR5 and the binding site of DNI-d2, there

would be at least another 12 nm (possible much more) added to distance between donor and acceptor. Is the proximity here close enough, that any signal from random collision would still be evident? I am concerned about interpretation of saturation TR-FRET using either a soluble (bulk solution) or lipophilic (membrane preferring) fluorescein to support the notion that random collision will produce a hyperbolic curve. This is indeed what they observe, but may be open to interpretation. They compare the hyperbolic curves (Fig S13) to linear plots (Fig 1D) assuming that the same ranges of concentration can be tested when transfecting a receptor and having this express at the cell-surface and adding a soluble fluorescent dye. If one examines fig S13, then it appears that if the first ~ 6 points were replotted without the highest concentrations this would be a good linear fit (as seen in Fig 1D for the receptor). One solution to the above difficulties in terms of interpretation would be to mutate CXCR4 to prevent dimerization and use this as a negative control. A dimeric structure for CXCR4 is published (PDB: 3OE0, doi.org/10.1126/science.1194396) from which it should be possible to design mutations that would preserve function/expression whilst blocking dimerization.

This second point raised by the Reviewer 2 dealt with a control for receptor oligomerization.. We used fluorescein and the fluorescent Membright and performed experiments with mGluR2 receptor component to prove that the signals obtained on CXCR4 and ACKR3 are specific and not due to random collisions. We believe that our data combined still clearly points towards the detection of the receptors in close proximity.

“Based on the published structure of mGluR5 and the binding site of DN1-d2”, Reviewer #2 suggested that the distance between donor and acceptor maybe increased. To our knowledge, DN1 is specific to mGluR2 receptor and do not bind to mGluR5. Moreover, it is really difficult to speculate about the DN1/mGluR2 conformation from another nanobody/receptor conformation since nanobody epitope are highly variable from one nanobody to another. The fact remains that Reviewer #2's hypothesis is perfectly valid and that we cannot rule out the possibility that the increased distance between the donor and acceptor is partly responsible for the absence of a significant signal between CXCR4 and mGlu2.

Finding a negative control for FRET experiments is always extremely difficult. The use of a class A receptor would certainly be more appropriate, but experience shows that any receptor that is sufficiently overexpressed will always end up interacting with the receptor of interest if it is addressed in the same cellular compartments.

Reviewer #2 suggested to include another control based on a mutant of CXCR4 that is unable to oligomerize. Unfortunately, while the idea is really attractive on paper, it would take a completely independent project to develop such a mutant. Indeed, based on the published CXCR4 dimer structure, various laboratories have tried unsuccessfully to find CXCR4 mutants which are unable to dimerize. This does not mean that such mutants cannot be found, but rather that a large-scale study is needed to identify and characterize them. Many control experiments will have to be performed to test these mutants before it can be concluded that they can even serve as a good control for our manuscript. For example, it is unknown which and how many mutations should be introduced into CXCR4 to prevent receptor oligomerization. There is also a chance that these mutations prevent receptor refolding or affect translocation of CXCR4 to the cell surface. Next, the binding of the nanobodies should be unaffected. And if, eventually, any signals are to be observed, would this then be a result from specific interactions or random collision?

Therefore, we agree with Reviewer 2 that such CXCR4 mutants are certainly very interesting but obtaining and characterizing them will be a research topic in itself and such a goal is not

achievable in the context of the present study and would be more fitting for its own dedicated manuscript.

Reviewer 2 have also suggested to move Figure S14 in the main manuscript. This would have been fine for us. However, throughout the manuscript, all control experiments are shown in Supplementary Information. For homogeneity, we think it is better to maintain this and keep Figure S14 there as well.

*There still appear to be ANOVAs that are performed where one group has been normalised to 100%, thus removing the variance of this group. This is not appropriate. Once again, I acknowledge the sloppy practice of other researchers in reporting SEM rather than SD and the use of different numbers of * to indicate different P values. With reference to SEM, I can only assume that when the authors walk in to their doctor and have their blood pressure taken, they must immediately call the ambulance when they find their diastolic is outside the population SEM..... for any well-trained biologist, an estimate of the population variance (i.e. SD) of any biological parameter is much more important than any measure of how accurately one has estimated the population mean (S.E.M.) - see references below. With respect to using different numbers of * to indicate different levels of statistical significance; this appears to indicate two things, 1, that the level of significance deemed to be appropriate has been decided post-hoc (a type of P hacking), and 2, that the authors believe that a lower P-value is related to a change in biological significance (untrue). I would suggest the following articles:*

Michel, M. C., Murphy, T. J., & Motulsky, H. J. (2020). New Author Guidelines for Displaying Data and Reporting Data Analysis and Statistical Methods in Experimental Biology. Molecular pharmacology, 97(1), 49–60. <https://doi.org/10.1124/mol.119.118927>

Halsey, L. G., Curran-Everett, D., Vowler, S. L., & Drummond, G. B. (2015). The fickle P value generates irreproducible results. Nature methods, 12(3), 179–185. <https://doi.org/10.1038/nmeth.3288>

Ronald L. Wasserstein & Nicole A. Lazar (2016) The ASA's Statement on p-Values: Context, Process, and Purpose, The American Statistician, 70:2, 129-133, DOI:10.1080/00031305.2016.1154108

Curran-Everett, D., & Benos, D. J. (2004). Guidelines for reporting statistics in journals published by the American Physiological Society. American journal of physiology. Regulatory, integrative and comparative physiology, 287(2), R247–R249. <https://doi.org/10.1152/ajpregu.00346.2004>

Curran-Everett, D., & Benos, D. J. (2007). Guidelines for reporting statistics in journals published by the American Physiological Society: the sequel. Advances in physiology education, 31(4), 295–298. <https://doi.org/10.1152/advan.00022.2007>

We apologize for the way we did the statistics for Figure S14 in the previous version of the manuscript. For the revised version, the statistics were compiled with the help of the biostatisticians of the institute in Montpellier. All results are now presented with standard deviation rather than SEM. In addition, they recommended to perform normalization and conduct a one-way ANOVA test using the multiple comparisons procedure. They also advised us to compare only the columns we wanted to compare and not performed a systematic comparison of all columns with each other. We followed their recommendations.

Also, with these new statistical tests, the FRET signals between VUN400 -Tb and VUN400-d2 on CXCR4 remains statistically different from that obtained in the presence of IT1t, whereas the FRET signals between VUN400-Tb and DN1-d2 is not.

It should be noted that under the experimental conditions we used, only 50% of CXCR4 dimers are labeled with both a donor and an acceptor (25% are labeled with two donors and 25% with two acceptors), whereas the hypothetical CXCR/mGluR2 dimers would all be labeled with a donor and an acceptor. This means that if we had been able to label all CXCR4 dimers with a donor and an acceptor, the difference between the two groups would have been even greater.

Taken together, even after revising our statistical analyses, the main conclusions of our study remain unchanged.

Rebuttal letter

Answers to the Reviewers

Reviewer #2 (Remarks to the Author):

I am pleased to see the additional specificity data provided in S11 panel C. This addresses the most critical issue regarding the ability of the results from transfected cell studies to be translated to endogenous settings with some degree of confidence.

I accept the difficulty with performing additional requested experiments. Having the most important specificity issue addressed is fine.

The authors appear to have changed the error bars in figure S11, S13 and S14 to SD. In their response they state that "All results are now presented with standard deviation rather than SEM" - this does not appear to be the case and all the figure legends (apart from those noted above) still state SEM. I don't know if this was perhaps a problem with not uploading the corrected figures and legends?

Indeed Reviewer 2 is right. We apologize for the mistakes. In the present version, in all the figures, data are expressed with SDs.

Statistics remain invalid for S14 due to normalization, non-parametric test such as Kruskal Wallis would address variance issue in control but would not fix the experimental design. Suggest doing statistics on non-normalised (raw data). Elsewhere statistics still have issues.

We performed a one-way Anova statistic test and raw data.